# Property Controllable Variational Autoencoder via Invertible Mutual Dependence

**Xiaojie Guo**
Department of IST
George Mason University
Fairfax, VA 22030, USA
xguo7@gmu.edu

**Yuanqi Du**
Department of CS
George Mason University
Fairfax, VA 22030, USA
ydu6@gmu.edu

**Liang Zhao***
Department of CS
Emory University
Atlanta, GA 30322, USA
liang.zhao@emory.edu

## Abstract

Deep generative models have made important progress towards modeling complex, high dimensional data. Their usefulness is nevertheless often limited by a lack of control over the generative process or a poor understanding of the latent representation. To overcome these issues, attention is now focused on discovering latent variables correlated to the data properties and manipulating these properties. This paper presents the Property-controllable VAE (PCVAE), where a new Bayesian model is proposed to inductively bias the latent representation using explicit data properties via novel group-wise and property-wise disentanglement terms. Each data property corresponds seamlessly to a latent variable, by enforcing invertible mutual dependence between them. This allows us to move along the learned latent dimensions to control specific properties of the generated data with great precision. Quantitative and qualitative evaluations confirm that the PCVAE outperforms the existing models by up to $28\%$ in capturing and $65\%$ in manipulating the desired properties. The code for the proposed PCVAE is available at:https://github.com/xguo7/PCVAE.

## 1 Introduction

Important progress has been made towards learning the underlying low-dimensional representation and generative process of complex high dimensional data such as images (Pu et al., 2016), natural languages (Bowman et al., 2016), chemical molecules (Kadurin et al., 2017; Guo et al., 2019) and geo-spatial data (Zhao, 2020) via deep generative models. In recent years, a surge of research has developed new ways to further enhance the disentanglement and independence of the latent dimensions, creating models with better robustness, improved interpretability, and greater generalizability with inductive bias (see Figures 1(a) and 1(b)) (Kingma et al., 2014; Kulkarni et al., 2015; Creswell et al., 2017) or without any bias (Higgins et al., 2017; Chen et al., 2018; Kumar et al., 2018). Although it is generally assumed that the complex data is generated from the latent representations, their latent dimensions are typically not associated with physical meaning and hence cannot reflect real data generation mechanisms such as the relationships between structural and functional characteristics. A critical problem that remains unsolved is how to best identify and enforce the correspondence between the learned latent dimensions and key aspects of the data, such as the bio-physical properties of a molecule. Knowing such properties is crucial for many applications that depend on being able to interpret and control the data generation process with the desired properties.

In an effort to achieve this, several researchers (Klys et al., 2018; Locatello et al., 2019b) have suggested methods that enforce a subset of latent dimensions correspond to targeted categorical properties, as shown in Figure 1(c). Though the initial results have been encouraging, critical challenges remain unsolved such as: (1) **Difficulty in handling continuous-valued properties**. The control imposed on data generation limits existing techniques to categorical (typically binary) properties, to enable tractable model inference and sufficient coverage of the data. However, continuous-valued properties (e.g., the scale and light level of images) are also common in real world data, while their model inference usually can be easily intractable. Also, many cases require to generate data

---

*Corresponding author: liang.zhao@emory.edu

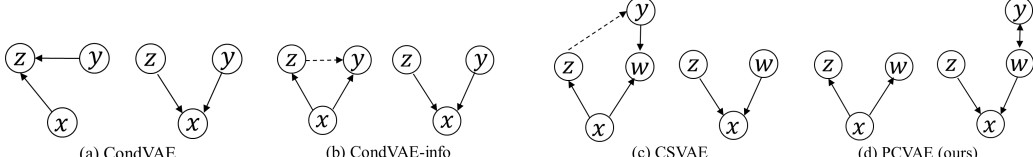

Figure 1: While most existing models (e.g., Sub-figures (a) (Kingma et al., 2014; Kulkarni et al., 2015) and (b) (Creswell et al., 2017)) do not explicitly learn the correspondence between latent dimensions and data properties, some recent work (Sub-figures (c) (Klys et al., 2018) and (d)) has started to explore this. The generative model (right) and its model inference (left) are shown in each sub-figure. Dotted arrows represent the enforcement of independence and double arrows represent the invertible dependence between two variables. $x$ refers to data, $z$ and $w$ refer to two subsets of latent variables, and $y$ refers to the properties.

with properties of which the values are unseen during training process. This cannot be achieved by conventional techniques such as conditional models without making strong assumption on the model distributions. (2) **Difficulty in efficiently enhancing mutual independence among latent variables relevant and irrelevant to the properties**. This problem requires to ensure that each property is only correlated to its corresponding latent variable(s) and independent of all the others. Directly enforcing such mutual independence inherently between all pairs of latent variables incurs quadratic number of optimization efforts. Hence an efficient way is imperative. (3) **Difficulty in capturing and controlling correlated properties**. It is feasible that several independent latent variables can capture multiple independent properties. But when the properties are correlated, they cannot be "one-on-one" mapped to corresponding independent latent variables anymore. However, correlated properties are commonly found in formatting a real world data.

To solve the above challenges, we propose a new model, Property-controllable VAE (PCVAE), where a new Bayesian model is proposed to inductively bias the latent representation using explicit data properties via novel group-wise and property-wise disentanglement terms. Each data property is seamlessly linked to the corresponding latent variable by innovatively enforcing an invertible mutual dependence between them, as shown in Figure 1(d). Hence, when generating data, the corresponding latent variables are manipulated to simultaneously control multiple desired properties without influencing the others. We have also further extended our model to handle inter-correlated properties. Our key contributions are summarized as follows:

- A new Bayesian model that inductively biases the latent representation using explicit real data properties is proposed. A variational inference strategy and inference model have been customized to ensure effective Bayesian inference.
- Group-wise and property-wise disentanglement terms are proposed to enhance the mutual independence among property, relevant and irrelevant latent variables .
- The invertible mutual dependence between property-latent variable pair is achieved by enforcing an invertibility constraint over a residual-based decoder.
- The quantitative and qualitative evaluation performed for this study revealed our PCVAE outperforms existing methods by up to $28\%$ in capturing and $65\%$ in manipulating the desired properties.

## 2 RELATED WORKS

**Disentanglement Representation Learning**. An important relevant area of research is disentangled representation learning (Alemi et al., 2017; Chen et al., 2018; Higgins et al., 2017; Kim & Mnih, 2018), which structures the latent space by minimizing the mutual information between all pairs of latent variables. The goal here is to learn representations that separate out the underlying explanatory factors that are responsible for variations in the data, as these have been shown to be relatively resilient with respect to the complex variants involved (Bengio et al., 2013; Ma et al., 2019; Guo et al., 2020), and thus can be used to enhance generalizability as well as improve robustness against adversarial attack. As noted by Locatello et al. (2019a), it is impossible for disentangled representation learning to capture the desired properties without supervision and inductive biases.

**Learning latent representations via supervision**. This ensures that the latent variables capture the desired properties though supervision, generally by directly defining properties as latent variables in the model (Locatello et al., 2019b). Unfortunately, apart from providing an explicit variable for the labelled property, this yields no other easily interpretable structures, such as discovering latent variables that are correlated to the properties, as the model proposed in the current study does. This

is also an issue with other methods of structuring latent space that have been explored, such as batching data according to labels (Kulkarni et al., 2015; Zhang et al., 2020) or using a discriminator network in a non-generative model (Lample et al., 2017). Some researchers addressed this problem by introducing the architecture bias through a two-way factored autoencoder and realize the supervision based on a pair-wise contrastive loss (Gyawali et al., 2019). Other researchers addressed this problem by linking latent variables with observed labels through adversarial learning (Creswell et al., 2017; Edwards & Storkey, 2015; Ganin et al., 2016; Mathieu et al., 2016). The most relevant work for our purpose is CSVAE (Klys et al., 2018), where a subset of latent variables are correlated with binary properties via an adversarial learning. All the above works can not handle multiple continuous-valued properties due to their strict assumptions on the distribution of properties.

**Data manipulation and generation**. Here, trained machine learning models are utilized to manipulate and generate data in a controllable way with the desired properties, which is especially useful for applications in the image domain. Several works have specifically considered transferring attributes in images, which is the same goal as that in the CASVE. These earlier works (Zhou et al., 2017; Xiao et al., 2017; 2018) all transfer attributes from a source image onto a target image. These models can only perform categorical attribute transformation between images (e.g., "splice the beard style of image A onto image B"), but only through interpolation between existing images. Once trained, our proposed model can generate an objects with any value of a certain property (either observed or unobserved during training) that can be encoded in the subset of latent variables.

## 3  PROPERTY CONTROLLABLE VAE

### 3.1  PROBLEM FORMULATION

Suppose we are given a dataset $\mathcal{D}$ where each data instance is $(x, y)$ with $x \in \mathbb{R}^n$ and $y = \{y_k \in \mathbb{R}\}_{k=1}^K$ to represent $K$ properties of interest of $x$. For example, if $x$ is a molecule, then we may have properties of interest, such as *cLogP* and *cLogS*. We assume that the data $(x, y)$ are generated by some random process from continuous latent random variables $(z, w)$. Each variable in $w$ controls one of the properties of interest in $y$, while the variables in $z$ control all the other aspects of $x$.

Our goal is to learn such a generative model involving $(x, y)$ and $(z, w)$, where the subset of variables (i.e., $z$) are disentangled from subset $w$, and the variables inside $w$ are disentangled from each other. Once this model has been learned, then we can expect different elements of variables in $w$ to control different properties of interest, which is a highly desirable goal for many data generation downstream tasks. For example, we may want to decrease the value of a specific property (e.g., protein energy) by changing the value of the corresponding element in $w$. It is also possible to directly set a desired property value (e.g., the mass of a molecule) and then generate the corresponding $x$ with this target value (i.e., a molecule with the target mass value).

### 3.2  OVERALL OBJECTIVE

In this section, we first introduce the Bayesian variational inference of PCVAE. Then we introduce the group-wise and property-wise disentanglement terms as part of the overall objective. Following this, an invertibility constraint is introduced to enforce mutual dependence between each property-latent variable pair. At last, PCVAE is extended to capture and control multiple correlated properties.

#### 3.2.1  BAYESIAN VARIATIONAL INFERENCE OF PCVAE

The goal in Section 3.1 requires us to not only model the dependence between $x$ and $(w, z)$ for latent representation learning and data generation, but also model the dependence between $y$ and $w$ for property manipulation. We propose to achieve this by maximizing a form of variational lower bound on the joint log likelihood $p(x, y)$ of our model. Given an approximate posterior $q(z, w|x, y)$, we can use the Jensen's equality to obtain the variational lower bound of $p(x, y)$ as:

$$\log p(x, y) = \log \mathbb{E}_{q(z,w|x,y)}[p(x, y, w, z)/q(z, w|x, y)]$$
$$\geq \mathbb{E}_{q(z,w|x,y)}[\log p(x, y, w, z)/q(z, w|x, y)]. \tag{1}$$

The joint likelihood $\log p(x, y, w, z)$ can be decomposed as $\log p(x, y|z, w) + \log p(z, w)$. We have two assumptions: (1) $w$ only encodes the information from $y$, namely, $x$ and $y$ are conditionally independent given $w$ (i.e., $x \perp y|w$); (2) $z$ is independent from $w$ and $y$, namely $z \perp w$

and $z \perp y$, which is equal to $y \perp z|w$ (see derivation in Appendix A.3). First, based on the two assumptions, we can get $x \perp y|(z, w)$ (see derivation in Appendix A.4). Thus, we have $\log p(x, y|z, w) = \log p(x|z, w) + \log p(y|z, w)$. Then, based on the assumption $y \perp z|w$, we can have $\log p(y|z, w) = \log p(y|w)$. Then we get $\log p(x, y|z, w) = \log p(x|z, w) + \log p(y|w)$. To explicitly represent the dependence between $x$ and $(z, w)$ as well as the dependence between $y$ and $w$, we can parameterize the joint log-likelihood as $\log p_{\theta,\gamma}(x, y, w, z)$ with $\theta$ and $\gamma$ as:

$$\log p_{\theta, \gamma}(x, y, w, z) = \log p_\theta(x|z, w) + \log p(z, w) + \log p_\gamma(y|w). \tag{2}$$

Given the condition that a parameterized $q_\phi(z, w|x, y) = q_\phi(z, w|x) = q_\phi(z|x)q_\phi(w|x)$ (since the information on $y$ is included in $x$), by taking Eq. 2 into the above variational lower bound term in Eq. 1, we obtain the negative part as an upper bound on $-\log p_{\theta, \gamma}(x, y)$ (as shown in the right sub-figure of Figure 1(d)):

$$\mathcal{L}_1 = -\mathbb{E}_{q_\phi(z, w|x)}[\log p_\theta(x|z, w)] - \mathbb{E}_{q_\phi(w|x)}[\log p_\gamma(y|w)] + D_{KL}(q_\phi(z, w|x)||p(z, w)) \tag{3}$$

This gives us the proposed Bayesian variational inference of PCVAE. The detailed derivation of Eq. 3 can be found in Appendix A.1. As there are $K$ properties of interest in $y$ which are assigned and disentangled by the latent variables in $w$, the second term in Eq. 3 can be detailed as $\sum_k^K \mathbb{E}_{q_\phi(w|x)}[\log p_\gamma(y_k|w_k)]$.

### 3.2.2 GROUP-WISE AND PROPERTY-WISE DISENTANGLEMENT

Considering that the above derivation is conditional on two requirements: (1) $z$ is independent from $w$ and $y$ and (2) the variables in $w$ are independent from each other, while in practice minimizing the above objective $\mathcal{L}_1$ will not imply that our model will satisfy these conditions. We therefore propose to further penalize the novel *Group-wise-* and *Property-wise Disentanglement* terms.

We first decompose the KL (Kullback-Leibler) divergence term in Eq. 3 as:

$$\mathbb{E}_{p(x)}[D_{KL}(q_\phi(z, w|x)||p(z, w))] = D_{KL}(q_\phi(z, w, x) \| q(z, w)p(x))$$
$$+ \underbrace{D_{KL}(q(z, w) \| \prod_{i,j} q(z_i)q(w_j))}_{\text{total correlation term}} + \sum_i D_{KL}(q(z_i) \| p(z_i)) + \sum_j D_{KL}(q(w_j) \| p(w_j)) \tag{4}$$

The second term in the right of the above equation is referred to as the total correlation (TC), as defined by Chen et al. (2018), which is one of many generalizations of mutual information to more than two random variables. The detailed derivation of this decomposition can be found in Appendix A.1. The penalty on this TC term forces the model to find statistically independent factors in the data distribution. A heavier penalty on this term induces a more disentangled representation among all variables in both $z$ and $w$, but as stated in our problem formulation, we only require that (1) variables in $w$ are disentangled to capture different properties, and (2) although $z$ is disentangled from $w$, the latent variables inside $z$ do not need to be disentangled from each other. Roughly enforcing the disentanglement between all pairs of latent variables in $w$ and $z$, as done by the existing TC term, can incur at least quadratic number of redundant optimization efforts and could lead to poor convergence. Thus, we further decompose and analyze the TC term as:

$$D_{KL}(q(z, w) \| \prod_{i,j} q(z_i)q(w_j)) = \underbrace{D_{KL}(q(z, w) \| q(z)q(w))}_{\text{group-wise disentanglement}} + \underbrace{D_{KL}(q(w) \| \prod_i q(w_i))}_{\text{property-wise disentanglement}}$$
$$+ D_{KL}(q(z) \| \prod_i q(z_i)). \tag{5}$$

The first term in the right part of above decomposition enforces the independence between the two subsets of latent variables $z$ and $w$, which we term *group-wise disentanglement*. The second term enforces the independences of variables inside $w$, ensuring that each latent variable can only capture the information of the single property assigned to it. We term this *property-wise disentanglement*. Imposing a heavy penalty on these two terms can satisfy the two requirements mentioned above. We can now obtain the second part for the objective of PCVAE by introducing the coefficient $\rho$ as:

$$\mathcal{L}_2 = D_{KL}(q(z, w)||q(z)q(w)) + \rho D_{KL}(q(w)|| \prod_i q(w_i)) \tag{6}$$

### 3.2.3 INVERTIBLE CONSTRAINT FOR PROPERTY CONTROL

As stated in the problem formulation, an important goal for our new model is to generate a data point $x$ that retains the original property value of a given property $y_k$ with great precision. More importantly, there should be no strict assumptions of parameters for $p(y_k)$ and $q(w_k|y_k)$. The most

straightforward way to do this is to model both the mutual dependence between $y_k$ and its relevant latent variable $w_k$, namely, $q(w_k|y_k)$ and $p(y_k|w_k)$, which, however, could incur double errors in this two-way mapping. To address it, we innovatively propose instead an invertible function that mathematically ensures the exact recovery of $w_k$ given $y_k$ based on the following deduction.

In the above objective, we only explicitly model the conditional distribution of $p_\gamma(y_k|w_k)$, hence, to achieve the precisely control of property via $z$ and $w$, which is necessary to generate $x$ with a certain property $y_k = m$, we need to maximize the probability that $y_k = m$ as follows:

$$x \sim p_\theta(x|z, w), \quad z, w \leftarrow \arg\max_{z \sim p(z), w \sim p(w)} p_\gamma(y_k = m|z, w), \tag{7}$$

which is equal to:

$$x \sim p_\theta(x|z, w), \ z \sim p(z), \ w_{j, j \neq k} \sim p(w_j), \quad w_k \leftarrow \arg\max_{w_k \sim p(w_k)} p_\gamma(y_k = m|w_k) \tag{8}$$

where $w_k$ can be determined as follows ($\mathcal{N}$ in the followings denotes Gaussian distribution):

$$
\begin{aligned}
w_k &\leftarrow \arg\max_{w_k \sim p(w_k)} p_\gamma(y_k = m|w_k) \\
&= w_k \leftarrow \arg\max_{w_k \sim p(w_k)} \log \mathcal{N}(y_k = m|f_k(w_k; \gamma), \sigma_k) \\
&= w_k \leftarrow \arg\max_{w_k \sim p(w_k)} -(m - f_k(w_k; \gamma))^2 = w_k \leftarrow f_k^{-1}(m)
\end{aligned}
\tag{9}
$$

Therefore, by learning an invertible function $f_k(w_k; \gamma)$ from $w_k$ to the expectation of $y_k$ to model $p_\gamma(y_k|w_k)$, we can easily achieve the desired precise control of the property. The above derivation are based on the assumption that $y$ is a continuous-value. It can also be extended into the situation when $y$ is discrete property, as detailed in Appendix A.2. To learn an invertible function $f_k(w_k; \gamma)$, we propose to leverage an invertible neural network. Inspired by the invertible ResNet (Behrmann et al., 2019), we decompose the function $f_k(w_k; \gamma)$ as $f_k(w_k; \gamma) = \bar{f}_k(w_k; \gamma) + w_k$. As proved by Behrmann et al. (2019), the sufficient condition that $f_k(w_k; \gamma)$ is invertible is that $Lip(\bar{f}_k) < 1$, where $Lip(\bar{f}_k)$ is the Lipschitz-constant of $\bar{f}_k(w_k; \gamma)$.

Thus, the overall objective of the proposed PCVAE is finally formalized as: $\min_{\theta, \phi, \gamma} \mathcal{L}_1 + \alpha \mathcal{L}_2$ with subject to $Lip(\bar{f}_k) < 1$ for all $k \in K$, where $\alpha$ is the coefficient parameter.

**Remark** (Monotonic relationship of property-latent variable pair). *Given the condition that $f_k(w_k; \gamma)$ is invertible and continuous ($Lip(\bar{f}_k)$ is less than 1), $f_k(w_k; \gamma)$ is thus a monotonic function. This is very important to increase (or decrease) the value of property $y_k$ by increasing (or decreasing) $w_k$, especially when the desired value of property is not available.*

### 3.2.4 GENERALIZATION OF HANDLING CORRELATED PROPERTIES

As stated in the third challenge in Section 1, there are usually several groups of properties involved in describing the data $x$ and each group has several properties. These different groups are independent, but the properties within the same group are correlated. Thus, we can further generalize the above objective framework to handle the correlated properties inside the same group.

The notation for $y_k$ is extended to $y_k = \{y_{j,k} \in \mathbb{R}\}_{j=1}^{M_k}$, signifying that there are $M_k$ correlated properties inside the $k$-th property group. The properties inside the same property set $y_k$ are correlated, while the different property sets (e.g. $y_p$ and $y_k$) are independent. Similarly, the notation for $w_k$ is extended as a group of latent variables to control the corresponding property set. For the properties inside the same group, we assume all depend on the same group of latent variables $w_k$ as

$$p(y_{j,k}|w_k) = \mathcal{N}(y_{j,k}|f_k(w_k; \gamma)[j], \sigma_{j,k})), \tag{10}$$

where $f_k(w_k; \gamma)[j]$ denotes the $j$-th element of the output of $f_k(w_k; \gamma)$. Thus, the second term in Eq. 3 can be generalized as $\sum_k^K \sum_j^{M_k} \mathbb{E}_{q_\phi(w|x)}[\log p_\gamma(y_{j,k}|w_k)]$.

### 3.3 NEURAL NETWORK ARCHITECTURE OF PCVAE

As shown in Figure 1(d), there is an encoder (left-hand side of Figure 1(d)) that models the distribution $q(z, w|x)$ and two decoders (right-hand side of Figure 1(d)) that model the distribution $p(y|w)$ and $p(x|z, w)$. To implement the encoder and decoders in the first objective (i.e., $\mathcal{L}_1$), we use Multi-perceptions (MLPs) , Convolution Neural Networks (CNNs) or Graph Neural Networks (GNNs) to represent the distributions over relevant random variables.

To implement the second part $\mathcal{L}_2$, it is necessary to calculate the group-wise and property-wise disentanglement terms. Noting that the calculation of the density $q(z)$, $q(w)$ and $q(w_i)$ in group-wise and property-wise disentanglement terms depends on the entire data space. As such, it is inaccessible to compute it exactly during training. Thus, as the same operation conducted by Chen et al. (2018), we utilize the Naïve Monte Carlo approximation based on a mini-batch of samples to underestimate $q(z)$, $q(w)$ and $q(w_k)$. The detailed operation is described in Appendix B.3.

To implement the invertible constraint and model the distribution of $p_\gamma(y_k|w_k)$, we utilize MLPs to model the function $\bar{f}_k(\cdot)$. Since the function $\bar{f}_k(\cdot)$ modeled by MLPs is a composition of contractive nonlinearities (e.g., ReLU, ELU, tanh) and linear mappings, based on the definition of Lipschitz-constant we have $Lip(\bar{f}_k) < 1$ if $\| W_l \|_2 < 1$ for $l \in L$, where $W_l$ refer to the weights of the $l$-th layer in $\bar{f}_k$, $\| \cdot \|_2$ denotes the spectral norm, and $L$ refers to the number of layer in the MLPs. To realize the above constraint on weights of neural networks, we propose to use the spectral normalization for each layer of MLPs, as introduced by Behrmann et al. (2019).

### 3.4 Precisely Property Controllable Generation

Our proposed model can be applied to an important downstream task, namely precisely property controllable generation. Given the value of a required property $y_k$, the goal of property controllable generation is to generate a data $x$ which holds the same value as this desired property. To achieve this, three steps are conducted: (1) infer the value of $w_k$ based on the well-trained neural network $\bar{f}_k(\cdot)$ and the given property $y_k$ via fixed-point iteration by following $w_k^{i+1} := y_k - \bar{f}_k(w_k^i; \gamma)$, where $w_k^i$ is the updated latent variable at the $i$-th iteration step and $w_k^0 = y_k$; (2) randomly sample the values of $z$ and the remaining variables in $w$ from their prior distributions (i.e., Gaussian distribution) to obtain all the latent variables; and (3) generate a data $x$ using the decoder based on the latent variables that are inferred from the previous two steps.

## 4 Experiment

This section reports the results of the qualitative and quantitative evaluation carried out to test the performance of the proposed model on two datasets in two domains, namely images and molecules. All experiments were conducted on a 64-bit machine with an NVIDIA GPU (GTX 1080 Ti, 11016 MHz, 11 GB GDDR5). The architectures and hyper-parameters can be found in Appendix B. The code for the proposed PCVAE is available at:https://github.com/xguo7/PCVAE.

### 4.1 Experiment Setup

#### 4.1.1 datasets

The **dSprites** dataset (Matthey et al., 2017) consists of 2D shapes procedurally generated from ground truth independent semantic factors. The factors that are explored as properties of data in this experiment are *scale*, and the $x$ and $y$ positions (mentioned as *x_pos* and *y_pos*) of a sprite. All possible combinations of these semantic factors are used for generating a total of $730k$ images, where $580k/146k$ is the training/testing set split. The **3Dshapes** dataset (Burgess & Kim, 2018) consists of 3D shapes procedurally generated from ground truth independent semantic factors. The factors that are explored as properties of data in this experiment are *wall hue*, *floor hue* and *scale*. All possible combinations of these semantic factors are used for generating a total of $480k$ images, where $390k/90k$ is the training/testing set split. The **QM9** dataset (Ramakrishnan et al., 2014) consists of $134k$ stable small organic molecules, where $120k/20k$ is the training/testing set split.

#### 4.1.2 Comparison Methods

In order to validate the superiority of our proposed model in capturing and manipulating the property during generation, we compare the performance of PCVAE to those achieved by three comparison models that are most relevant to our problem: (1) *semi-VAE* (Locatello et al., 2019b) is a semi-supervised model that enforces the value of each latent variable to be equal to the value of each property. Here we utilize all the labels for supervision for fairness; (2) *CSVAE* (Klys et al., 2018) is a VAE-based model that utilizes mutual information minimization to learn latent dimensions associated with only binary properties. Here we adjust this model to handle continuous property by assuming a Gaussian distribution; and (3) *PCVAE_tc* is a baseline model that

holds the same inference model and property controlling strategy as PCVAE, of which the proposed group-wise and property-wise disentanglement terms are replaced with the TC term, namely, $D_{KL}(q(z,w)|| \prod_{i,j} q(z_i)q(w_j))$, as proposed in $\beta$-*TCVAE* (Chen et al., 2018). It is used as an ablation study to validate the effectiveness of the proposed group-wise and property-wise disentanglement. (4) *PCVAE_nsp* is a baseline model that holds the same architecture and disentanglement terms as PCVAE except for the spectral normalization. It is used as an ablation study to validate the effectiveness of spectral normalization.

## 4.2 EVALUATION FOR DISENTANGLED LATENT VARIABLES

In this section, we explore (1) whether each variable $w_k$ successfully captures the information of the property that is assigned to it through supervision, and (2) whether the subset $z$ of latent variables is independent from the properties.

**Quantitative evaluation**. We calculate the normalized mutual information between each encoded latent variable $w_k$ and the property $y_k$ that is assigned to it, as well as the average mutual information between latent $z$ and each $y_k$. Figure 2 shows the mutual information heat map by each model on *dSprites*(results on QM9 datasets can be found in Appendix C). The element in the row of $z_{avg}$ denotes to the average of all the mutual information between $z$ and each property. In addition, we utilize the metric *avgMI*[1] proposed by Locatello et al. (2019b) to show an overall quantitative comparison between different methods, as shown in Table 1. The proposed PCVAE achieves the least avgMI of $0.257$, which demonstrate its strength in enforcing the relationship between $w_k$ and $y_k$. These results also validate the effectiveness and necessity of the proposed group-wise and property-wise disentanglement term, as PCVAE outperforms the baseline model PCVAE_tc on avgMI by around $28\%$. Though CSVAE shows an good performance in disentangling $z$ and $w$, its latent variables in $w$ turn in a poor performance for capturing the properties. Similar conclusions can also be drawn from the results on the 3Dshapes and QM9 dataset. For example, PCVAE outperforms the comparison models by about $16\%$ in capturing two independent properties *cLogP* and *Molweights*.

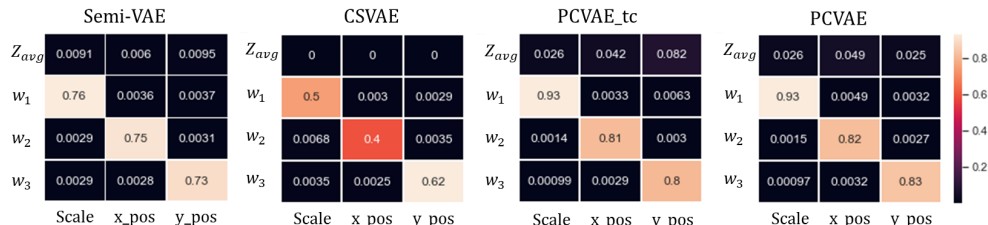

Figure 2: Heat-maps of the mutual information between latent variables and three properties by each model for the dSprites dataset. Data value in each cell denotes the normalized mutual information.

Table 1: The avgMI achieved by each model on the dSprites and QM9 datasets.

| Methods | dSpirites | 3Dshapes | QM9 |
|---|---|---|---|
| Semi-VAE | 0.439 | 0.115 | 1.413 |
| CSVAE | 0.868 | 1.348 | 1.411 |
| PCVAE_tc | 0.285 | 0.018 | 1.245 |
| PCVAE_nsp | 0.266 | 0.031 | 1.162 |
| PCVAE | **0.257** | **0.016** | **1.125** |

Table 2: MSE between the expected and actual property of the generated molecules (PCVAE (cor) denotes the extended model for correlated properties).

| Model | cLogP | Molweights | cLogS |
|---|---|---|---|
| Semi-VAE | 1.40 | 122.34 | N/A |
| CSVAE | 4.69 | 180.92 | N/A |
| PCVAE_tc | 5.02 | 131.15 | N/A |
| PCVAE_nsp | 1.81 | 176.94 | N/A |
| PCVAE | **1.29** | **87.62** | N/A |
| PCVAE (cor) | **1.33** | **53.49** | **1.15** |

**Qualitative evaluation**. We also qualitatively evaluate the dependence of each latent variable and its relevant property by visualizing the variation of the properties when traversing the priors of each latent variable. As shown in Figure 3, as the values of $w_1$, $w_2$ and $w_3$ change between $(-0.5, 0.5)$, the continuous variations of the assigned properties of *scale*, *x_position* and *y_position* of the generated images are clearly visible (as highlighted in red rectangle). The variables in $z = \{z_1, z_2, z_3\}$ have almost no influence on these properties, which validates the effectiveness of the group-wise disentanglement term. More evaluation results the other two datasets can be found in Appendix C.

---

[1] $avgMI = \| I(w,y) - E(k) \|_F^2$, where $k$ is the number of properties. $I(w,y)$ is mutual information matrix. The details can be found in Appendix B.4

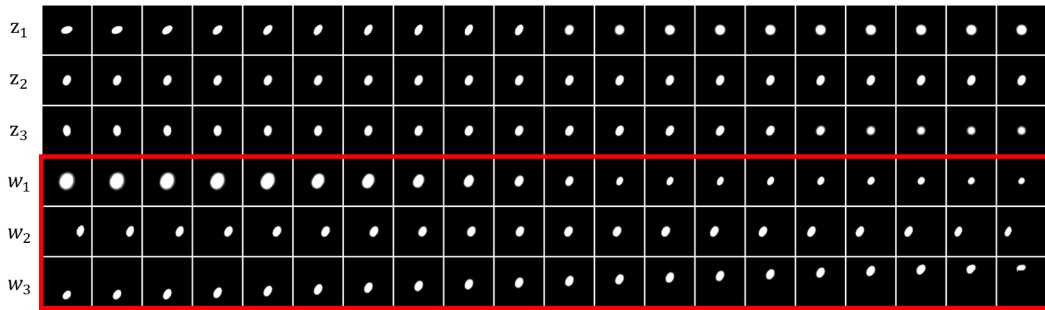

Figure 3: The generated images by PCVAE when traversing three latent variables in subset $z$ (upper 3 rows) and three latent variables in subset $w$ (bottom 3 rows) for the dSprites dataset.

### 4.3 EVALUATION FOR PROPERTY CONTROLLABLE GENERATION

In this section, we validate the performance of the property controllable generation. Specifically, given a predefined value of property $y_k$, the aim is to explore whether the proposed model could generate a data point $x$ with a property $y_k'$ that is the same as $y_k$.

**Qualitative evaluation**. For dSprites and 3Dshapes dataset, since we have no ground-truth method with which to calculate the property $y_k'$ of the generated images, we directly visualize the images that are generated given different values of property $y_k$. In Figure 4, each column contains four images generated given the same value of $y_k$ (here $y_k$ refers to the x_position property) but given the different values for the other two properties. The objects of the generated images in the same column clearly share the same *x_position*. Similar results can be observed in Figure 5, the wall hue or flooe hue of four images in the same column are the same given the same desired property. More visualizations for the other properties can be found in Appendix C.

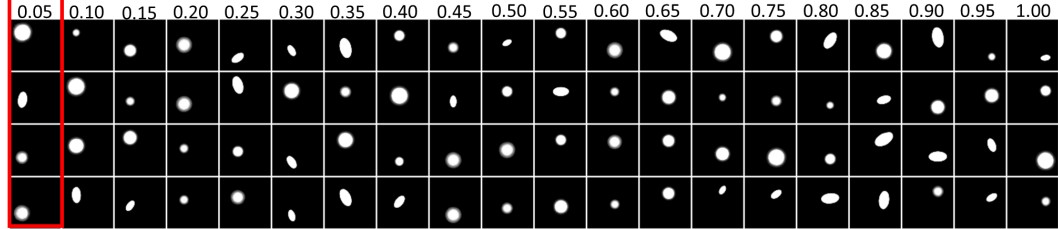

Figure 4: The generated images on dSprites dataset when traversing the desired value of property *X-position*. Each column of images are generated given the same value of the desired property.

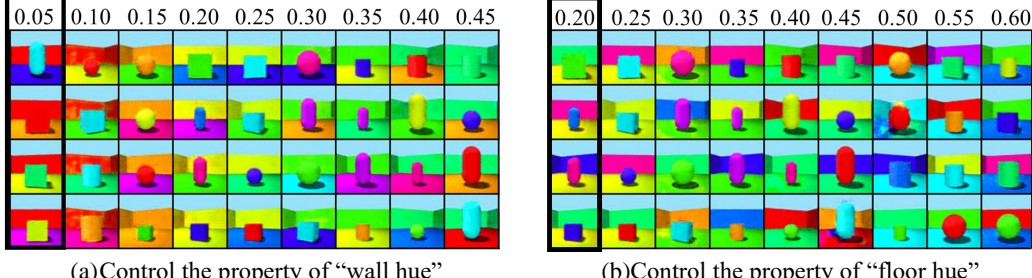

(a)Control the property of "wall hue"      (b)Control the property of "floor hue"

Figure 5: The generated images on 3Dshapes dataset when traversing the desired value of property (a) *wall hue* and (b)*floor hue*. Each column of images are generated given the same value of the desired property and random values of other properties.

**Quantitative evaluation**. For the QM9 dataset, since the properties can not be visualized directly from the molecule, we quantitatively measure the property controllable performance in terms of the **MSE** (mean squared error) between the actual property $y_k'$ of generated molecule and the desired property $y_k$, as shown in Table 2. The proposed PCVAE outperforms the other comparison models in successfully controlling *cLogP* and *Molweights* in molecule generation with a smaller **MSE** than that of the comparison methods by around $65.1\%$ and $40.5\%$ on average, respectively. This

demonstrate the superiority of PCVAE in precisely controlling the continuous-valued properties due to the effective invertible property prediction network. In addition, the obvious advantage of PC-VAE over PCVAE_tc demonstrate the effectiveness and necessity of the proposed group-wise and property-wise disentanglement term in precisely property controllable generation.

### 4.4 EVALUATION FOR HANDLING CORRELATED PROPERTIES

In this section, we access the ability of the proposed model to capture and control the correlated properties. The performance is tested on the QM9 molecule set for two tasks: property prediction and property controllable generation. Three properties are selected: *Molweights*, *cLogP* and *cLogS*. *Molweights* is independent from *cLogP* and *cLogS*, while *cLogP* and *cLogS* are inner-correlated.

First, we evaluate whether the subset of latent variables $w$ can power the ability of property prediction, which is a very important task for new compound design in drug discovery. Given an input molecule, the trained encoder (inference model) is used to get the relevant latent variable $w_k$, and then utilize the invertible function of $p(y_k|w_k)$ to predict the property $y_k$. Table 3 compares the performance of property prediction task on PCVAE and comparison models for uncorrelated properties, as well as the extended model (denoted as PCVAE(cor))

Table 3: Evaluation of the property prediction task in term of MSE between the predicted and real properties.

| Latent Variable | *Molweights* | *cLogP* | *cLogS* |
|---|---|---|---|
| semi-VAE | 975.18 | 1.44 | N/A |
| CSVAE | 63.78 | 1.21 | N/A |
| PCVAE_tc | 34.37 | 0.86 | N/A |
| PCVAE_nsp | **31.50** | **0.64** | N/A |
| PCVAE | 33.33 | 1.21 | N/A |
| PCVAE (cor) | **33.04** | **0.96** | **0.53** |

for correlated properties. Though PCVAE(cor) deals with a more difficult case than PCVAE, where an additional property *clogS* that is correlated with *cLogP* is included, PCVAE(cor) still successfully captures the information of the added property cLogS with ignorable influence on the prediction of cLogP and Molweight. Specifically, as shown in Table 3, regarding the prediction of independent properties, PCVAE outperforms semi-VAE and CSVAE with a smaller MSE of 33.33 in terms of *Molweights*. It can be also observed that the prediction results of PCVAE_nsp is better than PCVAE, which shows that enforcing both directions' dependence (i.e., $p(w|y)$ as well as $p(y|w)$ via spectral normalization) can introduce more errors than only modelling the dependence $p(y|w)$.

Next, we further explore the performance of the PCVAE(cor) to control the generation of the correlated properties. As shown in Table 2, PCVAE(cor) achieves the smallest MSE on all the properties. It demonstrate that adding the supervision of its correlated property *cLogS* does not influence the control of the property *cLogP*. This also demonstrates the effectiveness of the invertible function for handling multi-input and multi-output data. To test the necessities of the proposed PCVAE (cor), we also evaluate the performance of PCVAE and comparison models in dealing with correlated properties, of which the results could be found in Appendix C.4.

## 5 CONCLUSION

In this paper, we have proposed the PCVAE and its extended model, which learns a latent space that separates information correlated with the properties into a predefined subset of latent variables. To accomplish this, we first propose a novel Bayesian variational inference of PCVAE to jointly learn the distribution of data and properties followed by a novel group-wise and property-wise disentanglement term to deal with the complex dependency of subsets of latent variables. Then, we propose to enforce an invertible mutual dependence to allow the precise property controllable generation. At last, we demonstrate through quantitative and qualitative evaluations from three aspects that our proposed model achieves better performance than existing and baseline models. In future work, we plan to extend PCVAE to a semi-supervised setting, where some of the property labels are missing.

### ACKNOWLEDGMENTS

This work was supported by the National Science Foundation (NSF) Grant No. 1755850, No. 1841520, No. 2007716, No. 2007976, No. 1942594, No. 1907805, a Jeffress Memorial Trust Award, Amazon Research Award, NVIDIA GPU Grant, and Design Knowledge Company (subcontract number: 10827.002.120.04).

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

# A ADDITIONAL DERIVATIONS ABOUT METHODOLOGY

## A.1 DETAILED DERIVATION OF BAYESIAN VARIATIONAL INFERENCE OF PCVAE

Given an approximate posterior $q(z, w|x, y)$, we can use the Jensen's equality to obtain the variational lower bound of $p(x, y)$ as:

$$\log p(x, y) \geq \mathbb{E}_{q(z,w|x,y)}[\log p(x, y, w, z)/q(z, w|x, y)].$$

We have two assumptions: (1) $w$ only encode the information from $y$, namely, $x$ and $y$ are conditionally independent given $w$ (i.e., $x \perp y|w$); (2) $z$ is independent from $w$ and $y$, namely $z \perp w$ and $z \perp y$, which is equal to $y \perp z|w$ (see derivation in Appendix A.3). First, based on the two assumptions, we can get $x \perp y|(z, w)$ (see derivation in Appendix A.4). Thus, we have $\log p(x, y|z, w) = \log p(x|z, w) + \log p(y|z, w)$ . Then, based on the assumption $y \perp z|w$, we can have $\log p(y|z, w) = \log p(y|w)$. Then we get $\log p(x, y|z, w) = \log p(x|z, w) + \log p(y|w)$. To explicitly represent the dependence between $x$ and $(z, w)$ as well as the dependence between $y$ and $w$, we can parameterize the joint log-likelihood as $\log p_{\theta,\gamma}(x, y, w, z)$ with $\theta$ and $\gamma$ as:

$$\log p_{\theta,\gamma}(x, y, w, z) = \log p_\theta(x|z, w) + \log p(z, w) + \log p_\gamma(y|w). \tag{11}$$

Given the condition that a parameterized $q_\phi(z, w|x, y) = q_\phi(z, w|x) = q_\phi(z|x)q_\phi(w|x)$ (since the information on $y$ is included in $x$), by taking it into the above variational lower bound term, we obtain the negative part as an upper bound on $-\log p_{\theta,\gamma}(x, y)$ as:

$$\begin{aligned}
\mathcal{L}_1 =& \mathbb{E}_{q(z,w|x,y)}[\log p_\theta(x|z, w) - \log p(z, w) - \log p_\gamma(y|w) + \log q(z, w|x, y)] \\
&= -\mathbb{E}_{q(z,w|x,y)}[-\log p_\theta(x|z, w)] - \mathbb{E}_{q(z,w|x,y)}[\log p_\gamma(y|w)] \\
&+ \mathbb{E}_{q(z,w|x,y)}[\log q(z, w|x, y) - \log p(z, w)] \\
&= -\mathbb{E}_{q(z,w|x)}[-\log p_\theta(x|z, w)] - \mathbb{E}_{q(w|x)}[\log p_\gamma(y|w)] + \mathbb{E}_{q(z,w|x)}[\log \frac{q(z, w|x)}{p(z, w)}] \\
&= -\mathbb{E}_{q_\phi(z,w|x)}[\log p_\theta(x|z, w)] - \mathbb{E}_{q_\phi(w|x)}[\log p_\gamma(y|w)] + D_{KL}(q_\phi(z, w|x)||p(z, w))
\end{aligned} \tag{12}$$

Based on the above derivation of $D_{KL}(q_\phi(z, w|x)||p(z, w))$, we could further decompose it as:

$$\begin{aligned}
&D_{KL}(q_\phi(z, w|x)||p(z, w)) = \mathbb{E}_{q(z,w|x)}[\log \frac{q(z, w|x)}{p(z, w)}] \\
&= \mathbb{E}_{q(z,w|x)}[\log q(z, w|x) - \log p(z, w)] \\
&= \mathbb{E}_{q(z,w|x)}[\log \frac{q(z, w|x)}{q(z, w)} + \log \frac{q(z, w)}{\prod_{i,j} q(z_i)q(w_j))} + \log \frac{\prod_i q(z_i)}{\prod_i p(z_i)} + \log \frac{\prod_i q(w_i)}{\prod_i p(w_i)}].
\end{aligned}$$

Considering that $q(z, w) = \mathbb{E}_{p(x)}q(z, w|x)$, we can get:

$$\begin{aligned}
&\mathbb{E}_{p(x)}[D_{KL}(q_\phi(z, w|x)||p(z, w))] \\
&= \mathbb{E}_{p(x)}[\mathbb{E}_{q(z,w|x)}[\log \frac{q(z, w|x)}{q(z, w)} + \log \frac{q(z, w)}{\prod_{i,j} q(z_i)q(w_j))} + \log \frac{\prod_i q(z_i)}{\prod_i p(z_i)} + \log \frac{\prod_i q(w_i)}{\prod_i p(w_i)}]] \\
&= \mathbb{E}_{q(z,w,x)}[\log \frac{q(z, w, x)}{q(z, w)p(x)} + \mathbb{E}_{q(z,w)}[\log \frac{q(z, w)}{\prod_{i,j} q(z_i)q(w_j))}] \\
&+ \mathbb{E}_{q(z)}[\log \frac{\prod_i q(z_i)}{\prod_i p(z_i)}] + \mathbb{E}_{q(w)}[\log \frac{\prod_i q(w_i)}{\prod_i p(w_i)}]
\end{aligned} \tag{13}$$

$$\begin{aligned}
&= D_{KL}(q_\phi(z, w, x) \parallel q(z, w)p(x)) + D_{KL}(q(z, w) \parallel \textstyle\prod_{i,j} q(z_i)q(w_j)) \\
&+ \textstyle\sum_i D_{KL}(q(z_i) \parallel p(z_i)) + \sum_j D_{KL}(q(w_j) \parallel p(w_j))
\end{aligned} \tag{14}$$

## A.2 EXTENSION OF THE INVERTIBLE FUNCTION TO DISCRETE-VALUED PROPERTIES

Here we consider the situation when property $y$ is discrete-valued and we denote $y_k = \{0, 1\} \in \mathbb{R}^C$ as a one-hot vector here to represent its category and $C$ is the number of categories. In the overall

objective, we only explicitly model the conditional distribution of $p_\gamma(y_k|w_k)$, hence, to achieve the precisely control of property via $z$ and $w$, which is necessary to generate $x$ with a certain property $y_k$ that belong to the $m$th category, we need to maximize the probability that $y_k = M$ as follows:

$$x \sim p_\theta(x|z, w), \quad z, w \leftarrow \arg\max_{z \sim p(z), w \sim p(w)} p_\gamma(y_k = M|z, w), \quad (15)$$

where $M$ is also an one-hot vector with $M[m] = 1$. Then the above equation is equal to:

$$x \sim p_\theta(x|z, w), \ z \sim p(z), \ w_{j, j \neq k} \sim p(w_j), \quad w_k \leftarrow \arg\max_{w_k \sim p(w_k)} p_\gamma(y_k = M|w_k) \quad (16)$$

where $w_k$ can be determined as follows based on cross entropy objective (**Cat** in the followings denotes categorical distribution):

$$
\begin{aligned}
w_k &\leftarrow \arg\max_{w_k \sim p(w_k)} p_\gamma(y_k = M|w_k) \\
&= w_k \leftarrow \arg\max_{w_k \sim p(w_k)} \log \mathbf{Cat}(y_k = M|\nu_k(w_k; \gamma)) \\
&= w_k \leftarrow \arg\max_{w_k \sim p(w_k)} \log \frac{exp(\nu_k(w_k; \gamma)[m])}{\sum_j^C exp(\nu_k(w_k; \gamma)[j])} \\
&= w_k \leftarrow \nu_K^{-1}(M)
\end{aligned}
\quad (17)
$$

### A.3 DERIVATION PROCESS EXPLANATION 1

In this section, we proof that if $z \perp w$ and $z \perp y$, we can have $y \perp z|w$.
First, based on the Bayesian theory, we could have $p(y, z|w) = p(z|y, w)p(y|w) = p(y|z, w)p(z|w)$, namely,

$$p(z|y, w)p(y|w) = p(y|z, w)p(z|w). \quad (18)$$

Then, considering that $z \perp w$ and $z \perp y$, we can have $p(z|w) = p(z)$ and also $p(z|y, w) = p(z)$. Then the right and left sides of Equation 18 can be replaced as: $p(z)p(y|w) = p(y|z, w)p(z)$, and then we have $p(y|w) = p(y|z, w)$. Given $p(y|w) = p(y|z, w)$, both sides of the equations are multiplied by $p(z|w)$, and we have $p(z|w)p(y|w) = p(y|z, w)p(z|w) = p(y, z|w)$. Thus, $y \perp z|w$.

### A.4 DERIVATION PROCESS EXPLANATION 2

In this section, we proof that if $x \perp y|w$, $y \perp z$, and $z \perp w$, we can have $x \perp y|(w, z)$.

First, based on the Bayesian theory, we could have $p(x, y|w, z) = p(y|x, z, w)p(x|z, w) = p(x|y, z, w)p(y|z, w)$, namely,

$$p(y|x, z, w)p(x|z, w) = p(x|y, z, w)p(y|z, w) \quad (19)$$

Then, considering that $y \perp z$ and $z \perp w$ (namely $y \perp z|w$, as proved in Section A.3), as well as $y \perp x|w$, we can have $p(y|x, z, w) = p(y|w)$ and also $p(y|z, w) = p(y|w)$. Then the right and left sides of Equation 19 can be replaced as $p(y|w)p(x|z, w) = p(x|z, y, w)p(y|w)$, and then we have $p(x|z, w) = p(x|y, z, w)$. Thus, we get $x \perp y|(w, z)$.

## B ARCHITECTURE AND HYPER-PARAMETERS

### B.1 ARCHITECTURE AND HYPER-PARAMETERS FOR DSPRITS AND 3DSHAPES DATASET

Based on the description of the implementation of the proposed objective, there are three components, namely, *encoder_1* to model $p_\theta(z, w|x)$, *decoder_1* to model $p_\phi(x|z, w)$ and *decoder_2* to model $p_\gamma(w|y)$. When evaluate the dSprites data, the number of latent dimensions in $z$ is 3 and the number of latent dimensions in $w$ is also 3. The detailed architectures are shown in Table 4. The hyper-paramter used for training is detailed in Table 6.

### B.2 ARCHITECTURE AND HYPER-PARAMETERS FOR MOLECULE QM9 DATASET

The architecture used for evaluation on the QM9 dataset are totally borrowed from the work proposed by Liu et al. (2018b). A molecule is represented as a graph $G(X, A)$, where each atom is a

Table 4: Encoders and decoders architectures of PCVAE for dSprites dataset (Each layers is expressed in the format as $< kernel\_size >< layer\_type >< Num\_channel >< Activation\_function >< stride\_size >$. *FC* refers to the fully connected layers).

| Encoder_1 | Decoder_1 | Decoder_2 |
|---|---|---|
| Input: $X \in \mathbb{R}^{64 \times 64}$ | Input$[z, w] \in \mathbb{R}^6$ | Input:$w_k \in \mathbb{R}$ |
| $4 \times 4$ Conv.32 ReLU.stride 2 | FC.256 ReLU | FC.50 ReLU |
| $4 \times 4$ Conv.32 ReLU.stride 2 | FC.256 ReLU | Spectral_Norm Layer |
| $4 \times 4$ Conv.32 ReLU.stride 2 | FC.512 ReLU | FC.1 |
| $4 \times 4$ Conv.32 ReLU.stride 2 | $4 \times 4$ ConvTranspose.32 ReLU.stride 2 | N/A |
| FC.256 ReLU | $4 \times 4$ ConvTranspose.32 ReLU.stride 2 | N/A |
| FC.256 ReLU | $4 \times 4$ ConvTranspose.32 ReLU.stride 2 | N/A |
| FC.12 | $4 \times 4$ ConvTranspose.64 ReLU.stride 2 | N/A |

node and $X$ refers to the features for all nodes; each bond is an edge, where $A$ denotes to the adjacent matrix of the graph. Considering it is not the concentration of this paper, we briefly introduce the model and provide it architecture parameters in Table 5. We recommend the reader to the work (Liu et al., 2018b) for more details.

Table 5: Encoders and decoders architectures of PCVAE for QM9 dataset (Each layers is expressed in the format as $< kernel\_size >< layer\_type >< Num\_channel >< Activation\_function >< stride\_size >$. *FC* refers to the fully connected layers).

| Encoder_1 | Decoder_1 | Decoder_2 |
|---|---|---|
| Input: $G(X, A), X \in \mathbb{R}^9$ | Input$[z, w] \in \mathbb{R}^{100}, h\_type \in \mathbb{R}^{101}$ | Input:$w_k \in \mathbb{R}$ |
| FC.100 ReLU | FC.100 ReLU | FC.20 ReLU |
| GGNN.100 ReLU | GGNN.100 ReLU | Spectral_Norm Layer |
| GGNN.100 ReLU | GGNN.100 ReLU | FC.1 |
| FC.100 | FC.9 (node) FC.3 (edge) | N/A |

**Molecule Encoder and Decoder**. A *encoder_1* is constructed to model $p_\phi(z, w|x)$ based on a gated graph neural network (GGNN). As a result, by sampling from the modelled distribution, $(z, w)$ are obtained variables containing the graph representation vectors. The molecule *decoder_1* models the distribution $p_\theta(x|z, w)$ to generate the molecule graph $G$. The molecule *decoder_2* models the distribution $p_\theta(y|w)$ to predict the properties $y$. The process proceeds in an auto-regressive style. In each step a focus node is chosen to be visited, and then the edges are generated related to this focus node. The nodes are ordered by using the breadth-first traversal. The molecule decoder mainly contains three steps, namely *node initialization*, *node update* and *edge selection and labelling*.

**Node Initialization**. We first define $N$ as an upper bound on the number of nodes in the final generated graph. An initial state $h_i^{(t=0)}$ is assigned with each node $v_i$ in a set of initially unconnected nodes. Specifically, $h_i^{(t=0)}$ is the concatenation as $[(z, w), \tau_i]$, where $\tau_i$ is an one-hot vector indicating the atom type. $\tau_i$ is derived from $(z, w)$ by sampling from the softmax output of a learned mapping $\tau_i \sim f(Z_i)$. From these node-level states, we can calculate global representations $H(t)$, which is the average representation of nodes in the connected component at generation step $t$. In addition to $N$ working nodes, a special "stop node" is also initialized to a learned representation $h_{\text{end}}$ for managing algorithm termination detailed as below.

**Edge Selection and Labeling**    At each step $t$, a focus node $v_i$ is picked from the queue of nodes. Then an edges $e_{i,j}$ is selected from node $v_i$ to node $v_j$ with label $E_{i,j}$. Specifically, for each non-focus node $v_j$, we construct a feature vector $\eta_{i,j}^{(t)} = [h_i^{(t)}, h_j^{(t)}, d_{i,j}, H(t), H(0)]$, where $d_{i,j}$ is the graph distance (i.e. path) between two nodes $v_i$, $v_j$. We use these representations to produce a distribution over candidate edges as $p(e_{i,j}, E_{i,j}|\eta_{i,j}^{(t)}) = p(E_{i,j}|\eta_{i,j}^{(t)}, e_{i,j}) \cdot p(e_{i,j}|\eta_{i,j}^{(t)})$. The parameters of the distribution are calculated as softmax outputs from neural networks. New edges are sampled one by one from the above learned distributions. Any nodes that are connected to the graph for the first time during this edge selection are added to the node queue.

**Node Update**. Whenever we obtain a new graph $G^{(t+1)}$ at step $t$, the previous node states $h_i^{(t)}$ is discarded and a new node representations $h_i^{(t+1)}$ for each node is calculated by taking their (possibly changed) neighborhood into account. To this end, a standard gated graph neural network (GGNN) is utilized through $S$ steps, which is defined as a recurrent operation over messages $r_i^{(s)}$.

**Termination**. In the edge generation process of each node, the edges to a node $v_i$ is kept added until an edge to the stop node is selected. Then we move the focus from the node $v_i$, and regard $v_i$ as a "closed" node. The next focus node is then selected from the focus queue. In this way, a single connected component is grown in a breadth-first manner. The node and edge generations continue until the queue of nodes is empty.

Table 6: hyper-paramter used for training on dSprites and QM9 datasets

| Dataset | Learning_rate | Batch_size | $\alpha$ | $\rho$ | Num_iteration | c (spectral_norm) |
|---------|---------------|------------|----------|--------|---------------|-------------------|
| dSprites | 5e-4 | 64 | 3 | 1 | 20 | 0.97 |
| QM9 | 5e-4 | 32 | 1 | 6 | 100 | 1 |

### B.3 ESTIMATION OF GROUP-WISE AND PROPERTY-WISE DISENTANGLEMENT TERMS

To evaluate the density $q(z)$, $q(w)$ and $q(w_i)$ in the second loss $\mathcal{L}_2$, a naïve Monte Carlo approximation (Caflisch et al., 1998) is utilized for the estimation. We describe the operation by taking $q(z)$ as an example. A naïve Monte Carlo approximation based on a minibatch of samples from $p(n)$ ($n$ is the data sample index) is likely to underestimate $q(z)$. As stated by Chen et al. (2018), this can be intuitively seen by viewing $q(z)$ as a mixture distribution where the data index $n$ indicates the mixture component. With a randomly sampled component, $q(z|n)$ is close to 0, whereas $q(z|n)$ would be large if $n$ is the component that $z$ came from. So it is much better to sample this component and weight the probability appropriately. Thus, we can use a weighted version for estimating the function $\log q(z)$ during training. When provided with a mini-batch of samples $\{n_1, ..., n_M\}$, we can use the estimator as:

$$\mathbb{E}_{q(z)}[\log q(z)] \approx \frac{1}{M} \sum_{i=1}^{M} [\log \frac{1}{MN} \sum_{j=1}^{M} q(z(n_i)|n_i)], \tag{20}$$

where $z(n_i)$ is a sample from $q(z|n)$, and $N$ is the count of the whole samples in dataset. $M$ is the count of samples in a mini-batch.

### B.4 DETAILED DESCRIPTION OF AVGMI

To evaluate the performance of the disentangled representation learning of the inference model, we adopt the metric avgMI proposed by Locatello et al. (2019b). The goal of avgMI is to evaluate the whether each latent variable $w_k$ only capture the information of the relevant property $y_k$ and has nothing correlation with the other properties. We utilize the MI matrix (i.e. the matrix of pairwise mutual information between $w$ and $y$) to represent the overall disentanglement performance. The optimal MI matrix should be like an identity matrix with diagonal entries all 1 and other entries all 0, where the mutual information between each $w_k$ and $y_k$ is 1 and the MI between $w_k$ and other property $y_j$ is 0. Then avgMI score is calculated as the distance between the real MI matrix and the optimal MI matrix. The smaller the avgMI is, the better the performance are.

Each entry, namely, mutual information $\mathrm{MI}(w_i, y_j)$, in the mutual information matrix $I(x, y)$ is calculated as: $\mathrm{MI}(w_i, y_j) = \sum_{w_i} \sum_{y_j} [(p(w_i, y_j) \log \frac{p(w_i, y_j)}{p(w_i)p(y_j)})]$. Therefore, to empirically estimate $p(w_i)$, $p(y_j)$, and $p(w_i, y_j)$, we need to have $w$ and $y$ in the experiments. And as we know, we have the observations on $x$ and $y$, and $w$ is generated from $x$ by the encoder.

# C    ADDITIONAL EXPERIMENT RESULTS

## C.1    EVALUATION ON THE QUALITY OF GENERATION ON QM9

We evaluate the quality of the generated molecules on the QM9 dataset by three metrics: *Novelty* measures the fraction of generated molecules that are not in the training dataset; *Uniqueness* measures the fraction of generated molecules after and before removing duplicates; *Validity* measures the fraction of generated molecules that are chemically valid. The results of the evaluation are shown in Table 7. As shown in Table 7, the proposed PCVAE still achieve 100% valdity and 99.5%, which is desirable in the problem of controlling generation. We could also found that the proposed PC-VAE can have an influence on the uniqueness of the generated data, which may be explained by the supervision of the model. However, considering our focus is on data generation given the desired property, is not a critical issue, whereas the validity and novelty are still very high.

Table 7: Quantitative evaluation of the generated molecules.

| Method | Validity | Novelty | Uniqueness |
|---|---|---|---|
| GrammarVAE (Kusner et al., 2017) | 30.00% | 95.44% | 9.30% |
| GraphVAE (Li et al., 2018) | 61.00% | 85.00% | 40.90% |
| CGVAE (Liu et al., 2018a) | 100.00% | 96.33% | 98.03% |
| PCVAE_tc | 100.00% | 99.10% | 63.50% |
| PCVAE | 100.00% | 99.50% | 33.40% |

## C.2    EVALUATION ON DISENTANGLED LATENT VARIABLES

**Evaluation results on dSprites**. We provide the qualitative evaluation on the comparison experiments when traversing the values of latent variables in Fig. 6. As shown here, the latent variables $w$ learned by the baseline model PCVAE_tc could successfully capture the three properties, which validate the effectiveness of the proposed overall inference model. The latent variables $w_2$ and $w_3$ learned by CSVAE can capture the *x_pos* and *y_pos* properties, while $w_1$ fails in capture the *scale* property. Semi-VAE can capture the three properties but the quality of the generated images is very bad and biased.

**Evaluation results on 3Dshapes**. We provide the qualitative evaluation on the 3Dshapes when traversing the values of each latent variables $w$ in Fig. 6. As shown here, the latent variables $w$ learned by the proposed model PCVAE could successfully capture the three properties, object scale, wall hue and the floor hue in the images, which validate the effectiveness of the proposed overall inference model.

**Evaluation results on QM9 dataset**. We calculate the mutual information between each encoded latent variable $w_k$ and the property $y_k$ that assigned to it, as well as the average mutual information between latent $z$ and each $y_k$, as shown in Figure 8 for the results on molecule QM9 dataset. For this difficult task in the molecule domain which contains the implicit properties, the proposed PCVAE still shows significant advantage in capturing the *Mol_weight* and *cLogP* properties. We also qualitative evaluate the relationship of each latent variable and its relevant properties. We visualize the variation of the properties on QM9 datasets, when traversing on the priors of each latent variable, as shown in Figure 9. The variable $w_1$ and $w_2$ could successfully capture the properties *Molweight* and *cLogP*.

## C.3    EVALUATION FOR PRECISE PROPERTY CONTROL

**Evaluation on dSprites dataset**. For dsprits dataset, since we have no ground-truth method to calculatethe properties $y_k'$ of the image, we directly visualize the images that are generated given different properties $y_k$ on three comparison models. As shown in Figure 10, each column shows four images generated given the same value of $y_k$, but given the different values of other properties. It could be easily observed that the objects of the generated images in the same column have the same value of $y_k'$. We provide the visualization of *y_pos* in Fig.10.

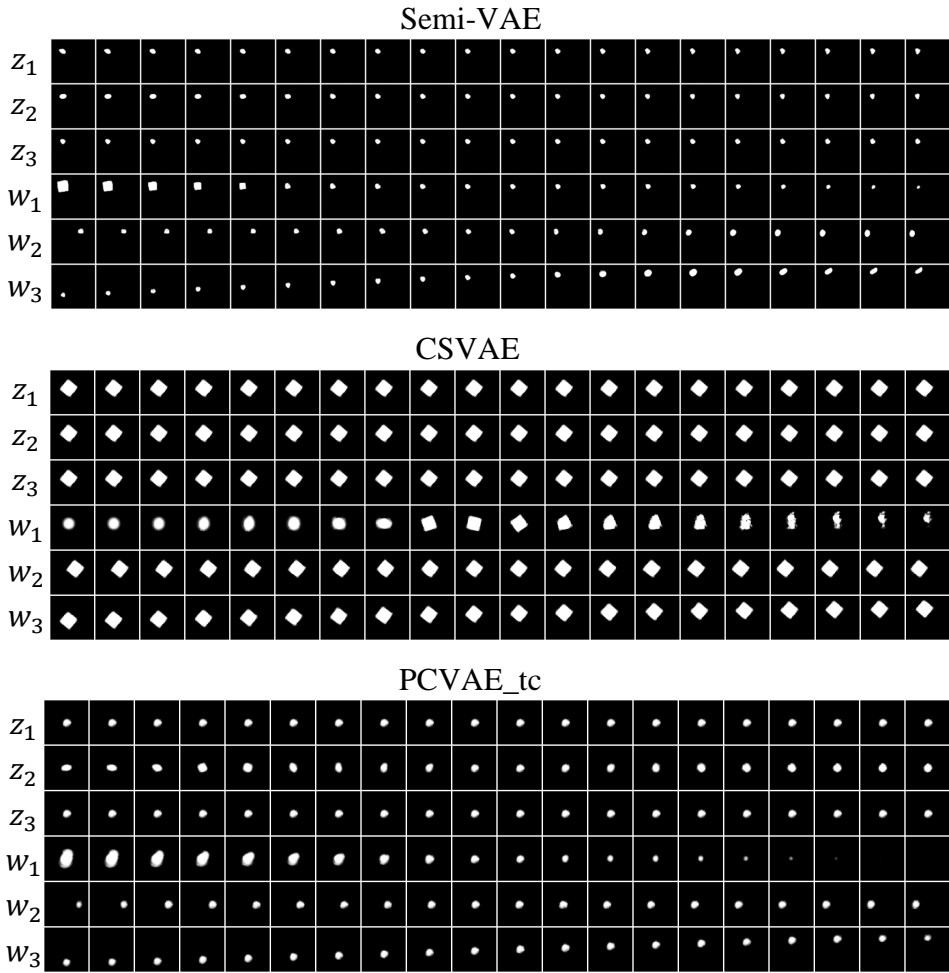

Figure 6: The generated images by comparison models when traversing on three latent variables in subset of latent $z$ (upper3 rows) and three latent variables in subset of latent $w$ (bottom 3 rows) for the Dsprits dataset

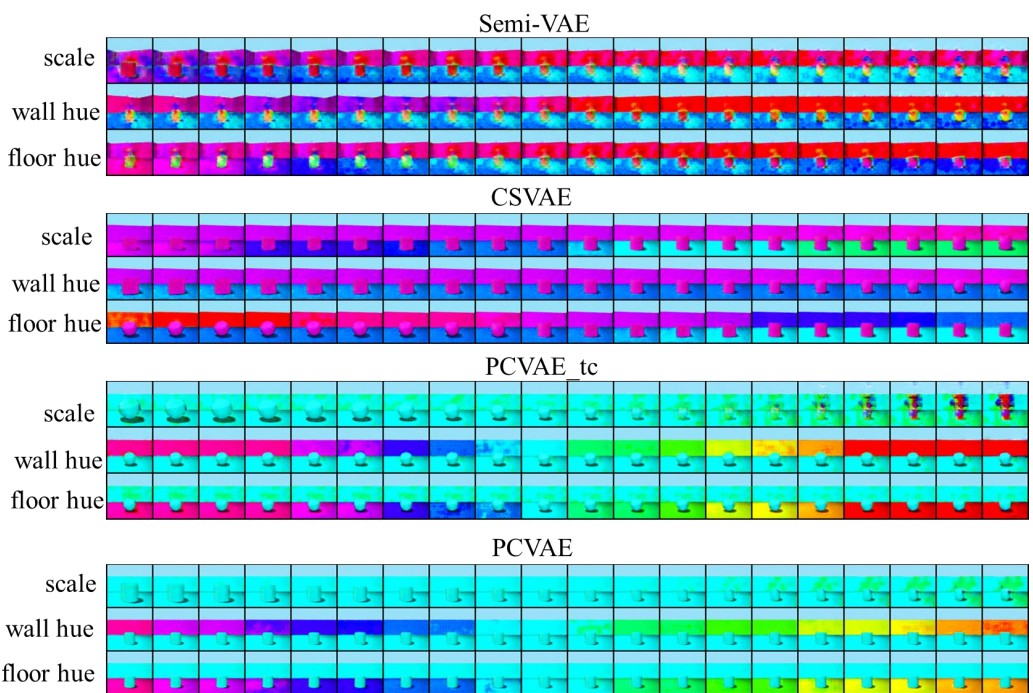

Figure 7: The generated images by PCVAE and comparison methods when traversing on three latent variables in subset of latent $w$ (bottom 3 rows) for the 3Dshapes dataset

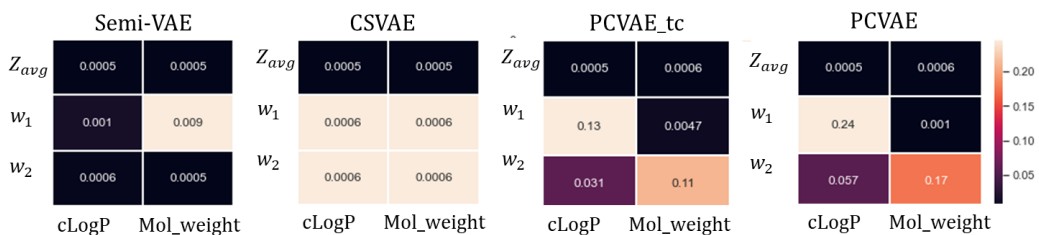

Figure 8: Comparison on Mutual Information between different latent variables on different properties on molecule QM9 dataset.

## C.4 EVALUATION ON THE NECESSARIES OF PCVAE(COR) FOR DEALING WITH CORRELATED PROPERTY

To validate the necessity of PCVAE (cor) for dealing with the correlated properties, we evaluated the performance of PCVAE and comparison models in dealing with correlated properties. As shown in Table 8, for generation task, the proposed PCVAE (cor) achieved much smaller MSE than those achieved by CSVAE and PCVAE by averagely around $72.9\%$, $52.5\%$ and $58.0\%$ on the control of

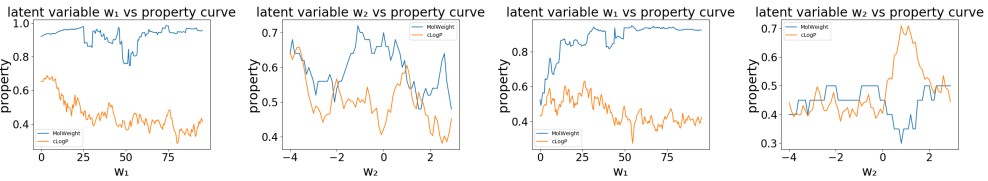

Figure 9: The properties of generated molecules when traversing on the corresponding latent variables in sub-space $w$ by ControlVAE_tc (left) and ControlVAE (right).

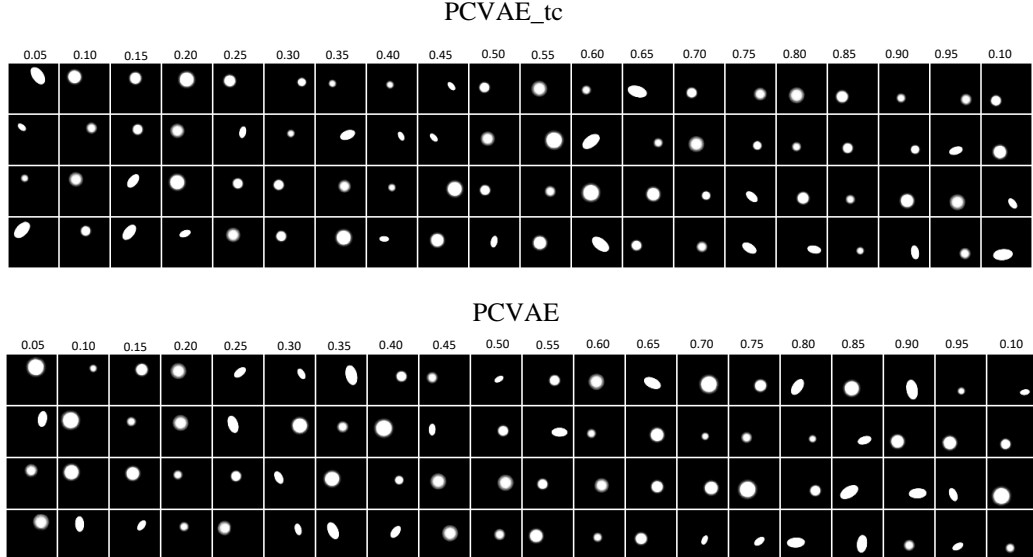

Figure 10: The generated images when traversing on the given desired property of Y-position. Each column of images share the same value of the desired property.

*Molweigt*, *cLogP* and *cLogS*, respectively. This validates that traditional disentangled-based VAE models cannot handle the controllable generation for correlated properties. For prediction task, the proposed PCVAE (cor) also achieved much smaller MSE than those achieved by CSVAE and PCVAE by averagely around $53.12\%$, $13.2\%$ and $22.1\%$ on the prediction of *Molweight*, *cLogP* and *cLogS* respectively. The bad performance on PCVAE and CSVAE in dealing with correlated properties is caused by the conflicts between the independence enforcement on latent variables $w$ and the dependence relationship enforcement on $w$ and the correlated properties $y$, which largely deteriorate the optimization of the whole model.

Table 8: Comparison of models in dealing with the correlated properties

| Model | Prediction task | | | Generation task | | |
|---|---|---|---|---|---|---|
| | *Molweight* | *cLogP* | *cLogS* | *Molweight* | *cLogP* | *cLogS* |
| CSVAE | 88.03 | 1.21 | 0.63 | 183.41 | 3.84 | 2.56 |
| PCVAE | 52.43 | 1.03 | 0.73 | 168.52 | 1.76 | 2.95 |
| PCVAE (cor) | **33.04** | **0.96** | **0.53** | **53.49** | **1.33** | **1.15** |

