# OpenReview forum: "Property Controllable Variational Autoencoder via Invertible Mutual Dependence"
_ICLR.cc/2021/Conference — ICLR 2021 Poster_

### Official Review · AnonReviewer2 · 2020-10-26
**Well motivated approach and good results on limited datasets**

**Rating:** 6
**Confidence:** 3

**Review:**

To encourage disentanglement in the latent space of a variational autoencoder (VAE), the authors propose to learn two sets of latent z and w: the dimensions of w are independent of each other and each dimension w_i maps to a known ground truth generating factor y_i. Latent z captures all the other factors. The well studied Total Correlation regularisation is used to enforce the independence of z and w, and the same is used to enforce the independence of the dimensions of w. Each dimension is learned to predict a corresponding ground truth factor. The key difference from the previous approach is the use of invertible and Lipschitz smooth mapping to learn monotonic mappings from w to y.

Pros:
+ The proposed regularization is shown to be useful for controlled manipulation for dsprites and QM9 dataset and improves upon comparable baselines.
+ Can be applied when the ground truth factors are continuous-valued.
+ Ability to handle the case where the ground truth factors are correlated.

Cons + Questions:
- Ablation studies on the use of spectral normalization will be helpful.
- The choice of Gaussian distribution for modeling the relationship between latent and ground truth factors could be elaborated upon.
- A more complex dataset at least in the case of natural images with large number of ground truth factors with correlated latent structure will be useful. In such cases how does the choice of dimensions of z and w impact the learning and their independence structure?


Typos:

Sec 2: earning -> learning

Sec 3.2.1 likely-hood -> likelihood

Fig 1(d) Y is missing from the model.

\phi is missing from RHS of eq 4.

---

> ### Author Response · Authors · 2020-11-14
> **Responses to Reviewer2**
>
> Thank you very much for your valuable time and comments. Please find our answers to your comments below. We have updated our paper based on suggestions from you. The summary of all the updates in the paper is listed in a separate comment on top of the page. If you have any further comments or suggestions on the updated version of our paper, we will be glad to improve on them. We also sincerely hope that the revised version and responses could help with an increase in the score.
>
> #### Summary of updates based on comments from Reviewer2:
>
> 1. We added the experiment on a new dataset: 3Dshapes.
> 2. We added the ablation study experiment for spectral normalization.
> 3. We have provided the codes of this paper in the supplemental materials.
> 4. We have added a simple variation of our method in handling discrete property in Appendix A. 2.
> 5. We have modified the typos and rephrased some hard-parsing sentences.
> ***
>
> **Comment #1:** *Ablation studies on the use of spectral normalization will be helpful.*
>
> **Response #1** Thanks for your suggestion. We have added the ablation study results of spectral normalization. The model without spectral normalization is denoted as “PCVAE_nsp”, as shown in Table 1, 2, and 3 on Page 7. As we expected, the results show that the spectral normalization has little influence on the disentanglement learning and prediction performance, but have a critical influence on property controllable generation, especially when dealing with the complex real-world dataset (i.e. QM9), as shown in Table 2. For example, the MSE of PCVAE_nsp is around 39% larger than that of PCVAE averagely, as shown below:
>
> | Model | cLogP | Molweights |
> |:----:|:----:|:----:|
> |semi-VAE|1.40|122.34|
> |CSVAE|4.69|180.92|
> |PCVAE_tc|5.02|131.15|
> |PCVAE_nsp|1.81|176.94|
> |PCVAE|**1.29**|**87.62**|
>
> More results can be found in the revised paper.
> ***
> **Comment #2:** *The choice of Gaussian distribution for modeling the relationship between latent and ground truth factors could be elaborated upon.*
>
> **Response #2:**
> 1. In Equation 9, the ground truth factors are assumed to be continuous-valued data. Thus, the prediction of continuous-valued $y$ from $w$ can be formalized as a regression problem, so Gaussian is widely used by default. Specifically, as we know for the regression problem, the prediction target is typically given by a deterministic function with additive Gaussian noise, which is equal to a Gaussian whose mean is the deterministic function (as specified in Section 3.3.2 (Page 156) in the book by Bishop [1]).
> 2. Our model is not limited to continuous-valued ground truth factors and Gaussian distribution. In the modified version, we have added the operation in handling the dependence between the discrete-valued factors and latent variables by formalizing it as a classification problem by utilizing the categorical distribution in Appendix A. 2.
>
> [1] Bishop, C. M. (2006). Pattern recognition and machine learning. springer.
> ***
> **Comment #3:** *“A more complex dataset at least in the case of natural images with large number of ground truth factors with correlated latent structure will be useful. In such cases how does the choice of dimensions of z and w impact the learning and their independence structure?*
>
> **Response #3:** Thanks for the suggestion.
> 1. We have added a new dataset: 3Dshapes in the revised version, which has the ground truth factors and is commonly utilized in the domain of disentangled representation learning.
> 2. It is difficult for us to build and evaluate a more complex natural dataset at the current stage. On one hand, the problem of “dealing with continuous correlated properties” is a very novel problem, thus there are few available datasets to test, especially the one with a large number of correlated factors. On the other hand, large efforts of annotations will be required when building the natural image dataset for supervision. Considering the intensive efforts needed and the significance in this topic, we are interested in exploring further down this line in future work.
>
> ***
> **Comment #4** *Typos issues*
>
> **Response #4:** Thanks for pointing out the typos. We have already modified all of them in the revised version except the third one. For the third typo raised by the reviewer, it is correct that there is no $y$ in the left-hand side of Fig 1(d). Since $y$ is only explicitly modeled in the generative model (right-hand-side), as consistent with the objective L1 in Equation 3 on Page 4.

---

### Official Review · AnonReviewer1 · 2020-10-27
**Nice experiments, but not self-contained enough to readers who are not familar with disentanglement literature.**

**Rating:** 6
**Confidence:** 4

**Review:**

**Update**

I appreciate the effort by the authors to clarify some of the issues, most of which are addressed in the rebuttal, so I will raise my score to 6.
I still feel like the $I(w, y)$ part needs to be dealt with a bit carefully, especially there is a invertible mapping between the two on the generative side. The simple graphical model seems like $w \leftarrow x \rightarrow y$, where left is encoder and right is data generation procedure.

**Summary**
The paper proposes property controllable VAE, with the aim to learn certain latent variables correlated to the property and are disentangled. This is done by variational inference + total-variation based disentanglement terms, with the aim to "control the properties".

**Strengths**
Empirical improvements over similar VAE approaches, such as Semi-VAE and CSVAE, showing the ability to learn and control the properties.

**Weakness**
 - The paper has several important details that are missing, and seems not self-contained enough to reproduce if the reader is not familiar with the disentanglement literature (see questions below).
 - I don't quite understand the motivation behind the method. If you want to control the value y (which is assumed to exist for all x), then why not replace w with y entirely (i.e. invertible network is identity) and operate directly on y? On inference side, predict $q(y | x)$, on generation side use empirical distribution (or some simple KDE) on y to get $p(y)$.
- Granted, it is possible that $y_i$s can be dependent, whereas you want $w_i$s to be independent; but it seems that Section 3.2.4 adds the assumption that $y_i$ are independent because you can group dependent variables?

**Questions**
- How are the total correlation objectives evaluated? For example, the explicit log-likelihood values for either $\log q(w)$ or $\log q(w_i)$ cannot be efficiently computed since that is the aggregate posterior? Is this implemented as an adversarial objective?
- What is the difference between PCVAE and PCVAE_tc (write down the equations)? Performance-wise they seem relatively similar.
- How does the invertible network work when y is discrete? If I made any error in w, then it maps to some y value outside of the sample space?
- How do you evaluate $I(w, y)$, and for which distribution is this defined? This confusion is because in reality there are two "worlds" concerning both variables (generative and inference directions) and because you have a invertible function between w and y (so it seems that optimizing MI is useless since you don't lose information anyways?).
- Why do we use avgMI as a metric? Do we expect the optimal MI value to be 1? If that is the case, then how do we get an avgMI of 1.004 for Semi-VAE in the first place?

---

> ### Author Response · Authors · 2020-11-14
> **Responses to Reviewer1 (Part I)**
>
> Thank you very much for your valuable time and comments. Please find our answers to your comments below. We have updated our paper based on suggestions from you. The summary of updates based on your comments in the paper is listed as follows. To help reproduce the proposed method, we have also provided the codes in the supplemental materials. If you have any further comments or suggestions on the updated version of our paper, we will be glad to improve on them. We hope that we have addressed your concerns and questions satisfactorily.
>
> #### Summary of the updates:
> 1. We have provided the codes of this paper in the supplemental materials.
> 2. We have mentioned the tool and methods to calculate the avgMI in Footnote 1 on Page 7.
> 3. We have added the detailed description of the calculation of the density $q(z)$ in Appendix B.3
> 4. We have added a simple variation of our method in handling discrete property in Appendix A.2.
> 5. We have added a detailed description of avgMI in Appendix B.4.
>
> ***
> **Comment #1:**  *I don't quite understand the motivation behind the method. If you want to control the value y (which is assumed to exist for all x), then why not replace w with y entirely (i.e. invertible network is identity) and operate directly on y? On inference side, predict, on generation side use empirical distribution (or some simple KDE) on y to get.*
>
> **Response #1:**
> 1. The alternative method mentioned by the reviewer is the same as semi-VAE [1] (one of the comparison models in the experiment), which directly defines properties as latent variables in the model, as discussed in the second paragraph of Section 2. However, this operation has two main drawbacks: (a) Difficult to assume the distribution of $y$. It requires to assume a prescribed distribution of $y$ (e.g., Gaussian in [1]) which is very difficult as most of the time the true distribution of the real-world properties in $y$ is unknown and too sophisticated to be predefined by simple distribution. However, our method can automatically approximate the sophisticated distribution on y by a learned mapping from $w$ to $y$. The mapping can be highly expressive leverage due to universal approximation theory, so very sophisticated distribution on $y$ can be learned without redefining; (b) Simply replacing $w$ with $y$ cannot handle the correlated properties. In many real-world applications the properties are correlated, hence if we discard w and directly use the properties in $y$ to control $x$ then, changing each property in $y$ will also lead to the change of the other correlated properties. So, the disentanglement among the (latent) variables, which is a widely desired goal in VAE, cannot be achieved.
> 2. Our model outperformed the alternative method mentioned by the reviewer (i.e., semi-VAE) in our experiment. The superiority of the proposed PCVAE over the semi-VAE is obviously observed in Tables 1 and 2. The limitation of semi-VAE is more obvious when dealing with complex real-world data. For example, regarding the control on Molweight property of molecules, the MSE score of semi-VAE is about 28% larger than the proposed PCVAE in Table 2.
>
> [1] Francesco Locatello, Michael Tschannen, Stefan Bauer, Gunnar Ratsch, Bernhard Sch ¨ olkopf, and ¨Olivier Bachem. Disentangling factors of variations using few labels. In International Conference on Learning Representations, 2019b.
> ***
> **Comment #2:** *Granted, it is possible that y_i(s) can be dependent, whereas you want w_i(s) to be independent; but it seems that Section 3.2.4 adds the assumption that y_i are independent because you can group dependent variables?*
>
> **Response #2:**
> 1. In the domain of disentangled representation learning, all the existing works conventionally focus on the independent ground-truth factors $y_i$(s) (i.e. properties) that to be captured by the latent variables $w_i$(s), which is the same as our PCVAE.
> 2. Beyond that, one of our contributions is that we are the first to break this conventional constrained assumption and extend our model to handle the dependent (correlated) properties, as stated in Section 3.2.4.
> ***
>
> **Question #1:** *How are the total correlation objectives evaluated? For example, the explicit log-likelihood values for either $log(q(w))$ or $log(q(w_i))$ cannot be efficiently computed since that is the aggregate posterior? *
>
> **Answer #1:**
> 1. We briefly mentioned this in the first paragraph in Section 3.3 in the original version. We utilize the Naïve Monte Carlo approximation based on a mini-batch of samples to underestimate $q(z)$, $q(w)$, and $q(w_k)$, which is commonly used (i.e. Chen et al. [2]).
> 2. Since the calculation of total correlation is not the focus of our paper, but just the utilization of existing well-recognized methods, we have added the detailed description in Appendix B.3 in the revised paper.
>
> [2] Ricky TQ Chen, et al. Isolating sources of disentanglement in variational autoencoders. NeurIPS, 2018.

---

> ### Author Response · Authors · 2020-11-14
> **Responses to Reviewer1 (Part II)**
>
> **Question #2:** *What is the difference between PCVAE and PCVAE_tc (write down the equations)? Performance-wise they seem relatively similar.*
>
> **Answer #2:**
> 1. As stated in the first paragraph of Section 4.1.2, the only difference is their disentanglement term in the overall objective. The disentanglement term in PCVAE_tc is the existing total correlation (TC) term proposed in β-TCVAE [2] (as in Equation 4), while the disentanglement term of PCVAE is the group-wise and property-wise disentanglement terms proposed in this paper (as in Equation 5). We added the equation for PCVAE_tc in the first paragraph of Section 4.1.2 in the revised version.  The comparison of their disentanglement terms is shown below:
>
> PCVAE:  $D_{KL}(q(z,w)||q(z)q(w))+\rho D_{KL}(q(w)||\prod\nolimits_{i} q(w_i))$
>
> PCVAE_tc: $D_{KL}(q(z,w)\parallel \prod\nolimits_{i,j} q(z_i)q(w_j))$
>
> 2. On one hand, the superiority of the performance of PCVAE_tc and PCVAE over the other methods indicates the effectiveness of the overall framework and invertible function in this paper. On the other hand, the observed superiority of PCVAE over PCVAE_tc in terms of the avgMI score in Table 1 and the MSE score in Table 2 validate the effectiveness of the proposed group-wise and property-wise disentanglement term.
>
> [2] Ricky TQ Chen, Xuechen Li, Roger B Grosse, and David K Duvenaud. Isolating sources of disentanglement in variational autoencoders. In Advances in Neural Information Processing Systems, pp. 2610–2620, 2018.
> ***
> **Question #3:** *How does the invertible network work when y is discrete? If I made any error in w, then it maps to some y value outside of the sample space?*
>
> **Answer #3:** That’s a good point.
> 1. Though this paper focuses on solving continuous-valued property issue which has not been handled by the existing models, the proposed invertible network can also deal with discrete property. The prediction of $y$ from $w$ can be formalized as a classification problem, where the probability of $y$ that belongs to any class is calculated based on an invertible function $f(w)$. The categorical distribution is assumed to model $p(y|w)$. We have added the extension part in handling discrete property in Appendix A.2.
> 2. Though some error may be made in $w$, the generated $y$ still falls into the sample spaces. This is because we assume a categorical distribution of $p(y|w)$ which ensures that the sampled output only belongs to the known categories.
> ***
>
> **Question #4:** *How do you evaluate I(w, y) and for which distribution is this defined? This confusion is because in reality there are two "worlds" concerning both variables (generative and inference directions) and because you have an invertible function between w and y (so it seems that optimizing MI is useless since you don't lose information anyways?)*
>
> **Answer #4:** Thanks for the comment.
> 1. The mutual information I(w,y) in the experiment is calculated based on the variables from the inference direction, namely the encoder part. In Section 4.2, I(w,y) is utilized to evaluate whether each encoded latent representation $w_k$ can capture the information of property $y_k$ of an input data $x$. Practically, we utilize the classic function in the publicity package “sklean” to calculate the normalized mutual information [3], where the input is a set of property $y_k$ and a set of encoded latent variable $w_k$ of the testing sample. We added the explanation in the revised version.
> 2. We are not optimizing MI directly in the proposed overall objective, as the reviewer mentioned, the low MI between a pair of relevant $w$ and $y$ can be achieved by the invertible function. More importantly, our optimization objective is that $z$ encodes little information of $y$ and the variables in $w$ are independent of each other, namely, each $w_k$ only has high MI with its relevant property $y_k$ but low MI with other properties.
>
> [3] https://scikit-learn.org/stable/modules/generated/sklearn.metrics.normalized_mutual_info_score.html
> ***

---

> ### Author Response · Authors · 2020-11-14
> **Responses to Reviewer1 (Part III)**
>
> **Question #5:** *Why do we use avgMI as a metric? Do we expect the optimal MI value to be 1? If that is the case, then how do we get an avgMI greater than 1 for Semi-VAE in the first place?*
>
> **Answer #5:**
> 1. avgMI is used in evaluating the disentanglement performance of the inference model in PCVAE, as proposed by Locatello et al [2]. We use avgMI to validate whether each variable $w_k$ captures only the information of the relevant property $y_k$ and has no correlation with the other properties.
> 2. avgMI is the distance between the real MI matrix and the optimal MI matrix, thus the value of AvgMI can be any real number no less than 0 (hence can be larger than 1).  Specifically, MI matrix refers to the matrix of pairwise mutual information between variables in $w$ and $y$. The optimal MI matrix should be like an identity matrix with diagonal entries all 1 and other entries all 0, indicating the MI between each $w_k$ and $y_k$ is 1 and the MI between $w_k$ and other property $y_j$ is 0. Then the smaller the avgMI is, the better the performance is.
> 3. We originally have a simple description in Footnote 1 on Page 6. We also added a more detailed description in Appendix B.4 in the revised version.
>
> [4] Francesco Locatello, Michael Tschannen, Stefan Bauer, Gunnar Ratsch, Bernhard Sch ¨ olkopf, and ¨Olivier Bachem. Disentangling factors of variations using few labels. In International Conference on Learning Representations, 2019b.

---

> ### Author Response · Authors · 2020-11-19
> **Further responses to Reviewer1**
>
> Dear Reviewer,
>
> Thanks for your updates. We are glad to further explain our graphical model and the metric $I(x,z)$ in the experiment as well as improve our paper presentation based on your suggestions.
> ***
> **Graphical model:**
>
> Our graphical model is illustrated in Figure 1(d) for the high-level intuitions about the proposed PCVAE. The encoder (inference part in left-hand side of Figure 1(d)) models the distribution $q(z,w|x)$. The decoder (generative part in right-hand side of Figure 1(d)) models the distribution $p(y|w)$ and $p(x|z,w)$. To fully make use of Figure 1 (d), we added the high-level description of our model in Section 3.3 as:
>
> *''As shown in Figure 1(d), there is an encoder (left-hand side of Figure 1(d)) that models the distribution $q(z,w|x)$ and two decoders (right-hand side of Figure 1(d)) that model the distribution $p(y|w)$ and $p(x|z,w)$."*
> ***
> **$I(x,y)$ in experiment evaluation:**
>
> 1. **About what is $I(x,y)$:** Each entry $I_{i,j}$ in the mutual information $I(w,y)$ is the mutual information between each latent variable $w_i$ and each property $y_j$ as: $MI(w_i,y_j)=\sum_{w_i}\sum_{y_j}[(p(w_i,y_j)\log\frac{p(w_i,y_j)}{p(w_i)p(y_j)})]$. Therefore, to empirically estimate $p(w_i)$, $p(y_j)$, and $p(w_i,y_j)$, we need to have $w$ and $y$ in the experiments. And as we know, we have the observations on $x$ and $y$, and $w$ is generated from $x$ by the encoder.
>
> 2. **About why $I(w,y)$ is a useful metric to evaluate our model's effectiveness:**  First, $w$ are directly encoded from $x$ via the encoder, which also can maintain the independence between $z$ and $w$, as well as the independence among different elements in $w$ via disentanglement regularization. Therefore, here the $w$ is not directly generated from $y$ using the invertible function. So, it is interesting to see if the $w$ generated by $x$ can still maintain its strong correlation with $y$, by also maintaining the independence between $z$ and $w$ as well as the independence among the elements of $w$. The experimental results validate that our model can have the best performance with the least avgMI of 0.257 on dSprites dataset, by outperforming semi-VAE (a model assumes a direct mapping which enforces $w=y$) by around 41.4%.
>
> 3. **About refinement of statements:** We refined the description in Appendix B.4 to make it clearer as:
> `*"Each entry, namely, mutual information $\mathrm{MI}(w_i,y_j)$, in the mutual information matrix $I(x,y)$ is calculated as: $\mathrm{MI}(w_i,y_j)=\sum_{w_i}\sum_{y_j}[(p(w_i,y_j)\log\frac{p(w_i,y_j)}{p(w_i)p(y_j)})]$. Therefore, to empirically estimate $p(w_i)$, $p(y_j)$, and $p(w_i,y_j)$, we need to have $w$ and $y$ in the experiments. And as we know, we have the observations on $x$ and $y$, and $w$ is generated from $x$ by the encoder. "*

---

### Official Review · AnonReviewer3 · 2020-10-28
**VAE for capturing explicit data properties.**

**Rating:** 6
**Confidence:** 4

**Review:**

The paper presents the new Property-controllable VAE (PCVAE) to inductively bias the latent representation to capture explicit data properties. Towards this, the paper proposes group-wise and property-wise disentanglement terms. The group-wise disentanglement term separates two subsets of the latent representation. In contrast, the property-wise disentanglement term promotes the disentanglement of an individual component of one of the subsets in the latent space. Furthermore, the model enforces individually disentangled latent subset to account for the given property of the dataset via invertible constraint. The presented work is evaluated on two datasets from two domains. Below I present some pros, cons, suggestions, and clarifying questions for the authors.

- The paper is clear to understand. The presented experimental studies are clear and demonstrate the efficacy of the model.  For instance, the precision with which the proposed model controls the property generation is impressive. Qualitatively this is demonstrated by Fig 4. Are these demonstrated on the train or the test set?

- For both group-wise and property-wise disentanglement terms, the TC term is considered. Can the authors clarify why they claim them to be novel? Can authors present training statistics to give insight into the model's optimization in relation to these up-weighted TC terms?

- dSprites is considered to be a relatively simpler dataset for disentanglement. If authors wanted a dataset with ground truth generative factors in the dataset, they could consider 3DShapes (Burgess & Kim, 2018), which would have more generative factors and is more challenging than dSprites. Also, 3DShapes is a widely considered dataset beside dSprites for disentanglement related study.

Burgess, C, & Kim, H. (2018). 3D Shapes Dataset. https://github.com/deepmind/3dshapes-dataset/.

- The literature review in the paper seems broad. I would encourage authors to discuss some recent approaches (e.g., Gyawali et al., 2019) that disentangle latent space into two subsets with one being related to certain property of the dataset.

Gyawali, P. K., Horacek, B. M., Sapp, J. L., & Wang, L. (2019). Sequential factorized autoencoder for localizing the origin of ventricular activation from 12-lead electrocardiograms. IEEE Transactions on Biomedical Engineering, 67(5), 1505-1516.

Minor comments:
- Please proofread your manuscript carefully to avoid grammatical errors. e.g., "latent presentation".

---

> ### Author Response · Authors · 2020-11-14
> **Responses to Reviewer3**
>
> Thank you very much for your valuable time and insightful comments. Please find our answers point by point below. We have updated our paper based on your suggestions. The summary of all the updates in the paper is listed in a separate comment on top of the webpage. If you have any further comments/suggestions on the updated version of our paper, we will be glad to improve on them. We also sincerely hope that the revised version and responses could be helpful with an increase in the final score.
>
> #### Summary of updated based on comments from Reviewer 3:
>
> 1. We have added the experiment on a new dataset: 3Dshapes.
> 2. We have provided the codes of this paper in the supplemental materials.
> 3. We have added additional related work analysis in the second paragraph in Section 2 on Page 2.
> 4. We have modified the typos and rephrased some hard-parsing sentences.
>
>
> ***
> **Comment #1:** *The paper is clear to understand…... Qualitatively this is demonstrated by Fig 4. Are these demonstrated on the train or the test set?*
>
> **Response #1:** Thanks for your comments. All the evaluation results displayed are measured based on the test set.
> ***
> **Comment #2:** *For both group-wise and property-wise disentanglement terms, the TC term is considered. Can the authors clarify why they claim them to be novel? Can authors present training statistics to give insight into the model's optimization in relation to these up-weighted TC terms?*
>
> **Response #2:** Thanks for this comment.
> 1. The novelty of our method is nontrivial. In our problem, we have three requirements in the disentanglement: variables in $w$ are disentangled (property-wise), $w$ is independent of $z$ (group-wise) and variables in $z$ are not required to be disentangled. The existing TC term cannot handle all of them. We propose new terms by decoupling and generalizing the existing TC term in a principled way. The proposed new terms can jointly handle group-wise and property-wise disentanglements but do not over-restrict on $q(z)$. Our model is more powerful since the trade-off among the group-wise and property-wise terms can all be adjustable, which is preferable in real applications.
> 2. Our model outperforms the TC term-based model (i.e. PVCAE_tc) in the ablation study experiments. For example, in the experiment on Q9 molecule dataset, the model with proposed terms (i.e., PCVAE) achieved a better performance than TC-term based model. This validated the advantage of the group-wise and property-wise disentanglement terms over the TC term.
>  ***
>
> **Comment #3:** *If authors wanted a dataset with ground truth generative factors in the dataset, they could consider 3DShapes (Burgess & Kim, 2018), which would have more generative factors and is more challenging than dSprites.*
>
> **Response #3:** Thanks for this suggestion. We have followed your suggestion and already added the evaluation results on the 3DShapes dataset, as shown in Table 1, and Figure 5 in the main text, and Figure 7 in the Appendix.
> ***
> **Comment #4:** *I would encourage authors to discuss some recent approaches (e.g., Gyawali et al., 2019) that disentangle latent space into two subsets with one being related to certain property of the dataset.*
>
> **Response #4:** Thanks for the recommendation. We have added the analysis of this work in the second paragraph in Section 2 as: “Some researchers have addressed this problem by introducing the architecture bias through a two-way factored autoencoder and realize the supervision based on a pair-wise contrastive loss (Gyawali et al. 2019)."
> ***
> **Minor comments #1:** *Please proofread your manuscript carefully to avoid grammatical errors. e.g., "latent presentation".
>
> **Response:** Thanks for this suggestion. We have fully proofread our paper and modified the typos in the revised version.

---

> > ### Comment · AnonReviewer3 · 2020-11-21
> > **Thanks for the update.**
> >
> > Thanks for the detailed response and the care with which you addressed my concerns. I appreciate it!

---

### Official Review · AnonReviewer4 · 2020-10-28
**Another tweak on disentanglement in VAEs with continous properties for supervision**

**Rating:** 6
**Confidence:** 3

**Review:**

**Property controllable variational autoencoder via invertible mutual dependence**

The paper proposes an approach for learning disentangled latent representation in the VAEs where parts of the latent space should correspond to observed properties of interest of the data so that these could be controlled at generation. They replace the total correlation regularization term of Chen2018 with more appropriate group- and property-wise KL terms promoting independence between the *relevant* (w) and the *remaining* (z) part of the latent space and inner independence between dimensions of the w part of the space (dropping the term promoting independence in the z part.) More importantly, they link the data property y with the latent space w by a learned mapping.

The is another contribution to the rather extensive literature on disentanglement of latent representation of VAEs. It builds on previous papers in the area, mainly Chen2018 and Klys2018. The change in the ELBO as compared to Chen2018 (TC vs group/property-wise) is well explained and motivated though seems rather trivial.
The introduction of the invertible mapping between y and w is in my view more interesting and important. Though somewhat contrived at first reading (some improvements in the text could help, see comments), it seems to correspond well to what the model shall achieve.
This  as is also demonstrated in the experimental section focusing mainly on the dSprites and QM9. The paper would benefit from a more extensive experimental evaluation (another standard dataset such as CelebA) though I understand this may be difficult to do given the deadlines and computational resources and therefore do not consider as critical.
The paper is generally well written though there are places which are difficult to parse and comprehend and suggest last minute drafting. These shall be made clear for the final version (see comments below).
The problem of correlated properties is addressed by a rather trivial extension of the basic model and requires the user to provide to the model groups of correlated properties. I feel this deserves a lot more attention and search for better solutions is a possible direction for future work.

I recommend to accept the paper as I find it interesting for the community, addressing a lively area of research via a new approach which, though not dramatically innovative, seems to yield the desired results. I do have a few comments as to the clarity of the statements in the paper which I hope will be addressed by the authors during the rebuttal period and in the final version.

*Comments / questions:*
* I find calling the mapping between y and w *invertible* as misleading. The mapping $w \to y$ is defined as stochastic via the learned generative distribution $p(y | w)$. Each w thus corresponds to multiple possible values of $y$.

* eq (2): please state clearly (in equations) the independence and conditional independence assumptions leading to this result

* eq (4) holds only in expectation over $p(x)$, right? ($E_{p(x)}$ in front of the DKL in the left hand side.)

* p4 before eq (5): "Roughly disentangling all pairs of latent variables without emphasis could lead to poor convergence and incur exponential number of pairs among properties and latent variables for such enforcement." What do you mean? Please explain / elaborate / rephrase.

* p4, "... no strict assumptions of parameters for $p(y_k)$ and $q(w_k |y_k)$". $y$ does not exist in the inference model so what does $q(w_k |y_k)$ refer to?

* p4 "The most straightforward way to do this is to model both the mutual mapping between $y_k$ and its relevant latent variable $w_k$." Both mappings $y_k \to w_k$ and its inverse $w_k \to y_k$? Or what do you mean?

* p5, para5 "Thus, we utilize the Naı̈ve Monte Carlo approximation based on a mini-batch
of samples to **underestimate** $q(z), q(w)$ and $q(w_k)$, as described by Chen et al. (2018)" Underestimate? What do you mean?

* p6: you say you use the normalized mutual information between $w_k$ and $y_k$. Please explain how you define this and how you calculate in practice.

* p7, fig3 "... when traversing three latent variables in subset z ...". There are only 3 latent variables z? If more, how did you decide which shall be traversed?

* p8, tab3: How do you predict the property $y_k$ from a molecule? You first infer w and z and then use the mean of p(y | w) as prediction? Or you sample y from p(y | w)? Please elaborate.

* p7, tab2: Do I read correctly that MSE of cLogP is bigger from the PCVAE (corr) then from the standard PCVAE? What does this say about the advantages of PCVAE (corr)?

* experiments over QM9: are all your evaluation metrics calculated only over valid molecules? Usually, generative models for molecules are evaluated using the validity, unicity and novelty scores. It would be instructive to complement your results with these.


*Minor comments / questions*
* Please provide references to models displayed in Fig1(a)-(c).

* You use the term *correletad* a lot. I'm guessing here you mean any sort of dependence, not just linear?

* p2: "Directly enforcing such mutual independence inherently between all pairs of latent variables incurs exponential number of pairs among properties and latent variables for such enforcement." Exponential number of pairs? What do you mean?

* p2, para3: "...  as these have been shown to be relatively resilient with respect to the complex variants involved (Bengio et al., 2013)." Complex variants? What do you mean?

* p3, para2:  $y = {y_k \in R}_{k=1}^K$. Is this the same as $y \in R^K$?


*Typos or phrasing improvements needed:*

* p1, para1: "**Knowing such properties is crucial** for many applications that depend on being able to interpret the data and control the data generation **to yield the desired properties**." ???

* p2. para1: "Also, many cases require to generate data with properties of which the values **are?** unseen during training process."

* p3 just before eq (2): $\log P_{θ,γ} (x, y, w, z)$ (capital P?)

* p5, para3: "As stated in the third challenge in Section 1, there are usually several groups of properties involved in **formatting**? data x ..."

* p6, para5: "... each encoded latent variable $w_k$ and the property $y_k$ **that is?** assigned to it ..."

* p7, caption Tab2: "...  (PCVAE (cor) denotes the **extensive**? model for correlated propertie"

---

> ### Author Response · Authors · 2020-11-14
> **Responses to Reviewer4 (Part I)**
>
> Thank you very much for your detailed summarization and insightful comments. Please find our answers to your comments/questions below. We have updated our paper based on your suggestions. The summary of updates in the paper are listed in a separate comment on top of the webpage. If you have any further comments/suggestions on the updated version of our paper, we will be glad to improve on them. We also sincerely hope that the revised version and responses could help with an increase in the score.
>
> **A summary of updates based on comments from Reviewer4:**
> 1. We have provided the codes of this paper in the supplemental materials.
> 2. We added the experiment on the new dataset: 3Dshapes.
> 3. We have modified the typos and rephrased some hard-parsing sentences.
> 4. We have added the references in the caption of Figure 1.
> 5. We have added a clarification about invertible dependence in the first line after Equation 9 on Page 5.
> 6. We added the detailed derivation of Equation 2 on Page 4.
> 7. We added the annotation of expectation in Eq (4) and detailed derivation of Eq (4) in Appendix A.1.
> 8. We have added the detailed description of the calculation of the density $q(z)$ in Appendix B.3
> 9. We have added the evaluation results on validity, unicity, and novelty of QM9 dataset in Appendix C.1
>
> ***
> **Comment #1**: *The paper would benefit from a more extensive experimental evaluation (another standard dataset such as CelebA)*`
>
> **Response #1:** That’s a good point. We have added the experiment of a new dataset named “3Dshapes” in the revised paper. The results are shown in Table 1 and Figure 5. We do not select CelebA since it does not contain the annotated ground-truth labels for supervision.
> ***
> **Comment #2:** *I find calling the mapping between $y$ and $w$ invertible as misleading. The mapping is defined as stochastic via the learned generative distribution. Each w thus corresponds to multiple possible values of y.*`
>
> **Response #2:** Thanks for this suggestion.
>
> 1. In fact, it is the mapping function $f_k(w_k)$ that is invertible. The stochasticity is added over function $f_k(w_k)$, via $\mathcal{N}(y_k=m|f_k(w_k;\gamma),\sigma_k)$, where $\sigma_k$ means standard deviation, as shown in Equation 9. We have added the clarification in the revised paper in the first line after Equation 9 on Page 5.
> 2. To eliminate the misleading issue, we modified two places in the paper that mentions “invertible mapping” and change them into the “invertible dependence” in the revised version.
> ***
> **Comment #3:**  *eq (2): please state clearly (in equations) the independence and conditional independence assumptions leading to this result.*
>
> **Response #3:** We have added the detailed derivation as well as the independence assumptions in equations in the revised version before Equation 2 on Page 4 as:
> “The joint likelihood $\log p(x,y,w,z)$ can be decomposed as $\log p(x,y|z,w)+\log p(z,w)$. First, by assuming $w$ only encode the information from $y$, namely, $x$ and $y$ are conditionally independent given $w$ (i.e.,$x\perp y|w$), we can have $\log p(x,y|z,w)=\log p(x|z,w)+\log p(y|z,w)$. Next. by assuming that $z$ is independent from $w$ and $y$, namely $z \perp w$ and $z \perp y$, we can have $\log p(y|z,w)=\log p(y|w)$. Then we get $\log p(x,y|z,w)=\log p(x|z,w)+\log p(y|w)$. To explicitly represent the dependence between $x$ and $(z,w)$ as well as the dependence between $y$ and $w$, we can parameterize the joint log-likelihood as $\log p_{\theta,\gamma}(x,y, w,z)$ with $\theta$ and $\gamma$ as: $\log p_{\theta,\gamma}(x,y, w,z)=\log  p_{\theta}(x|z,w)+\log p(z,w)+\log p_{\gamma}(y|w).$
> ***
> **Comment #4:** *eq (4) holds only in expectation over p(x) right? (E_{p(x)} in front of the DKL in the left-hand side.)*
>
> **Response #4:** Yes, you are right. Thanks for pointing this. We added the annotation of expectation in Eq (4) in the revised version. To make the derivation process clearer, we have also added the detailed derivation of Eq (3) and Eq (4) in Appendix A.1.
>  ***
> **Comment #5:** *p4 before eq (5): ‘Roughly disentangling all pairs of latent variables without emphasis could lead to poor convergence and incur exponential number of pairs among properties and latent variables for such enforcement.’ What do you mean? Please explain / elaborate / rephrase.*
>
> **Response #5:** (1) We revised the sentence as: “Roughly enforcing the disentanglement between all pairs of latent variables in $w$ and $z$, as done by the existing TC term, can incur at least quadratic number of redundant optimization efforts and could lead to poor convergence.”  (2) To explain it, suppose we have $N$ variables in $z$ and $M$ variables in $w$. The traditional TC term enforces disentanglement among $(M+N)^2$ pairs of variables, while only $(N^2+N)$ pairs of variables are necessarily needed to be disentangled in our problem. Thus, the TC term includes lots of redundant enforcement components which adds much burden on the optimization.

---

> > ### Comment · AnonReviewer4 · 2020-11-18
> > **Thanks for all the clarification, a couple more points ...**
> >
> > Comment #3:
> > * To be able to write $log p(x, y∣z, w) = log p(x∣z, w) + log p(y∣z, w)$ formally you need not only $x \perp y | w$ but also $x \perp y | z$, don't you?
> > * $log p(y∣z, w) = log p(y∣w)$ is not a result of either $z \perp w$ nor $z \perp y$, is it? It would only be true if $y \perp z | w$.
> >
> > Comment #5: Ok, clear, though you haven't actually updated this in the text of the paper (I'm guessing just minor mishap)
> >
> > Comment #12: Comparisons of PCVAE_corr with the other methods in Table 2 and Table 3 are kind of pointless as they give NA for cLogS for the other methods. If I understand correctly, there is nothing to prevent the other methods to operate over both cLogP and cLogS though these are correlated. As the tables are, we can only assume (or believe you) they would not do too well, but really?
> > (Aslo, I just spotted that PCVAE_nsp is overall the best in Table3. Any comments?)

---

> > > ### Author Response · Authors · 2020-11-20
> > > **Further responses on Comments #3 and #5**
> > >
> > > Dear Reviewer:
> > >
> > > We really appreciate your feedback. Followings are our responses to comments #3 and #5. We are still in the process of adding the comparison experiment you mentioned in Comment #12 but will provide the results soon.
> > > ***
> > > **Comment #3.1:**  $\log p(y|z,w)=\log p(y|w)$ can not got from $z\perp w$ and $z\perp y$. We need to have: $y\perp z |w$.
> > >
> > > **Answer #3.1:**  Actually, $z\perp w$ and $z\perp y$ can lead to $\log p(y|z,w)=\log p(y|w)$, which also leads to $y\perp z |w$. The detailed derivation process is shown as below:
> > >
> > > First, based on the Bayesian theory, we have $p(y,z|w)=p(z|y,w)p(y|w)=p(y|z,w)p(z|w)$, namely, we have:
> > >
> > > $p(z|y,w)p(y|w)=p(y|z,w)p(z|w)..........(1)$
> > >
> > > Then, considering that $z \perp w$ and $z\perp y$, we can have $p(z|w)=p(z)$ and also $p(z|y,w)=p(z)$. Then the right and left sides of Equation (1) can be replaced as $p(z)p(y|w)=p(y|z,w)p(z)$, and then we have $p(y|w)=p(y|z,w)$.
> > > This also says that $z\perp w$ and $z\perp y$ are equal to $y\perp z |w$. This is because if $z\perp w$ and $z\perp y$, we have $p(y|w)=p(y|z,w)$, and then we have $p(z|w)p(y|w)=p(z|w)p(y|z,w)=p(z,y|w)$ (this part is added as Appendix A.3).
> > > ***
> > >
> > > **Comment #3.2:** To be able to write $\log p(x,y|z,w)=\log p(x|z,w) +\log p(y|z,w)$, you need not only $x\perp y|w$, but also $x\perp y|z$.
> > >
> > > **Answer #3.2:** Our assumptions for this decomposition do not only include $x\perp y|w$, but also $y\perp z$ and $z \perp w$. We refined the misleading description after Equation 1 in the revised paper as:
> > >
> > > *"We have two assumptions: (1) $w$ only encodes the information from $y$, namely, $x$ and $y$ are conditionally independent given $w$ (i.e., $x\perp y|w$); (2) $z$ is independent from $w$ and $y$, namely $z \perp w$ and $z \perp y$, which is equal to $y\perp z |w$ (see derivation in Appendix A.3).  First, based on the two assumptions, we can get $x\perp y |(z,w)$ (see derivation in Appendix A.4). Thus, we have $\log p(x,y|z,w)=\log p(x|z,w)+\log p(y|z,w)$. Second, based on the assumption $y\perp z |w$, we can have $\log p(y|z,w)=\log p(y|w)$. Thus, we finally have $\log p(x,y|z,w)=\log p(x|z,w)+\log p(y|w)$."*
> > >
> > > Appendix A.4 is provided as below for easy reference:
> > >
> > > *"First, based on the Bayesian theory, we can have $p(x,y|w,z)=p(y|x,z,w)p(x|z,w)=p(x|y,z,w)p(y|z,w)$, namely, we have:
> > >
> > > $p(y|x,z,w)p(x|z,w)=p(x|y,z,w)p(y|z,w).........(2)$
> > >
> > > Then, considering that $y\perp z |w$, as well as $y\perp x|w$, we can have $p(y|x,z,w)=p(y|w)$ and also $p(y|z,w)=p(y|w)$ (as proved in Appendix A.3). Then the right and left sides of Equation (2) can be replaced as $p(y|w)p(x|z,w)=p(x|z,y,w)p(y|w)$, and then we have $p(x|z,w)=p(x|y,z,w)$. Thus, we get $x\perp y|(w,z)$."*
> > > ***
> > > **Comment #5:** you haven't actually updated this in the text of the paper (I'm guessing just minor mishap)
> > >
> > > **Answer #5:** Thanks for your reminder. We have updated the new version of the text.

---

> > > > ### Comment · AnonReviewer4 · 2020-11-23
> > > > **Thanks**
> > > >
> > > > Thank you for all the clarifications, I hope these will be useful for all readers of the paper. I appreciate your effort.

---

> > > ### Author Response · Authors · 2020-11-22
> > > **Further responses on Comment #12**
> > >
> > > **Comment #12:**  Comparisons of PCVAE_corr with the other methods in Table 2 and Table 3 are kind of pointless as they give NA for cLogS for the other methods. If I understand correctly, there is nothing to prevent the other methods to operate over both cLogP and cLogS though these are correlated. As the tables are, we can only assume (or believe you) they would not do too well, but really? (Also, I just spotted that PCVAE_nsp is overall the best in Table3. Any comments?)
> > >
> > > **Answer #12:**
> > > Thanks for your valuable feedback.
> > >
> > > 1. Since typical disentangled representation learning requires the properties to be independent, we treat this setting as the main task for our experiment. For correlated properties, theoretically, the comparison models and proposed PCVAE are not reasonable methods since they assume and enforce $w$ to be independent and each $w$ corresponds to a property $y$, where properties $y$ should also be independent.
> > >
> > > 2. We followed your suggestions to experimentally test the necessity of PCVAE (cor) and have provided the results in Appendix C.4 (due to the time limit, we only test CSVAE and PCVAE, which is enough to validate the necessity of PCVAE (cor)). As you suggested, since the critical credit of this paper is about property controllable generation, we are interested in exploring further down the problem of correlated properties in future work.
> > >
> > > Appendix C.4 is provided as below:
> > >
> > >
> > > *"To validate the necessity of PCVAE (cor) for dealing with the correlated properties, we evaluated the performance of PCVAE and comparison models in dealing with correlated properties. As shown in Table 8, for the generation task, the proposed PCVAE (cor) achieved much smaller MSE than those achieved by CSVAE and PCVAE by averagely around 72.9%, 52.5%, and 58.0% on the control of Molweigt, cLogP, and cLogS, respectively. This validates that traditional disentangled-based VAE models cannot handle the controllable generation for correlated properties. For the prediction task, the proposed PCVAE (cor) also achieved much smaller MSE than those achieved by CSVAE and PCVAE by averagely around 53.12%, 13.2%, and 22.1% on the prediction of Molweight, cLogP, and cLogS respectively. The bad performance on PCVAE and CSVAE in dealing with correlated properties is caused by the conflicts between the independence enforcement on latent variables $w$ and the dependence relationship enforcement on $w$ and the correlated properties $y$, which largely deteriorate the optimization of the whole model."*
> > >
> > > Table 8: Comparison of models in dealing with the correlated properties in terms of MSE for prediction and generation task
> > >
> > > |model | Molweight (prediction) |cLogP (prediction) |cLogS (prediction)|Molweight (generation) |cLogP (generation) |cLogS (generation)|
> > > |:-----:|:-----: |:-----:|:-----:|:-----:|:-----:|:-----:|
> > > |CSVAE |88.03 |1.21|0.63|183.41|3.84|2.56|
> > > |PCVAE | 52.43 |1.03|0.73|168.52| 1.76|2.95|
> > > |PCVAE (cor) |**33.04**|**0.96**|**0.53**|**53.49**|**1.33**|**1.15**|
> > >
> > >
> > >
> > >
> > >
> > > 3. PCVAE_nsp is the baseline model which is similar to PCVAE except being without the spectral normalization (as stated in Section 4.1.2). PCVAE_nsp has a smaller prediction MSE than PCVAE since the spectral normalization on PCVAE can influence the learning of $p(y|w)$, which is critical to prediction performance. This is because enforcing both directions’ dependence (i.e., $p(w|y)$ as well as $p(y|w)$) via spectral normalization is much complex and thus introduce more errors, than learning the dependence $p(y|w)$ alone. However, without spectral normalization, the property control of the generated data is impossible, as shown in Table 2. We added a sentence of comment regarding this in the second paragraph of Section 4.4 as:
> > >
> > > *“ It can be also observed that the prediction results of PCVAE_nsp are better than PCVAE, which shows that enforcing both directions’ dependence (i.e., $p(w|y)$ as well as $p(y|w)$) via spectral normalization can introduce more errors than only modeling the dependence $p(y|w)$.”*

---

> > > > ### Comment · AnonReviewer4 · 2020-11-23
> > > > **Thanks**
> > > >
> > > > Good it works the way expected. Seeing the numbers helps.

---

> ### Author Response · Authors · 2020-11-14
> **Responses to Reviewer4 (Part II)**
>
> **Comment #6:** *p4, "... no strict assumptions of parameters for $p(y_k)$ and $q(w_k/y_k)$ ".  y does not exist in the inference model so what does $q(w_k/y_k)$ refer to?*
>
> **Response #6:**
> 1. This conditional distribution $q(w_k/y_k)$ models the dependence between $w_k$ and $y_k$. Specifically, given a data $x$ with property $y_k$, we could infer its relevant latent variable $w_k$.
> 2. Though we do not explicitly model the conditional distribution $q(w_k/y_k)$ in the inference model, but $y_k$ is the property of $x$ and $ q(w_k/x)$ exists in the inference model, thus $q(w_k/y_k)$ does exist naturally. Actually we can get $ q(w_k/x)=q(w_k/y_k)$ as all the information in $x$ other than $y_k$ are independent to $w_k$.
> ***
> **Comment #7:** *p4 "The most straightforward way to do this is to model both the mutual mapping between $y_k$ and its relevant latent variable $w_k$ ." Both mappings $y_k$-> $w_k$ and its inverse $w_k$->$y_k$ ? Or what do you mean?*
>
> **Response #7:** Yes, “both” here means the two directions of dependence. As suggested by your first comment, we have rephrased the “mapping” and elaborate this in the revised version as: “The most straightforward way to do this is to model the mutual dependence between $y_k$ and its relevant latent variable $w_k$, namely, $q(w_k/y_k)$ and $p(y_k/w_k)$…”.
> ***
> **Comment #8** *p5, para5 "Thus, we utilize the Naı̈ve Monte Carlo approximation based on a mini-batch of samples to underestimate q(z), q(w) and q(w_i), as described by Chen et al. (2018)" Underestimate? What do you mean?*
>
> **Response #8:** Thanks for this comment.
> 1. We added a simple explanation in the revised version as:
> “To implement the second part $\mathcal{L}_2$, it is necessary to calculate the group-wise and property-wise disentanglement terms. Noting that the calculation of the densities $q(z)$, $q(w)$, and $q(w_i)$ in group-wise and property-wise disentanglement terms depends on the entire data space, which is not accessible. Thus, as the same operation conducted by Chen et al [1], we utilize the Naïve Monte Carlo approximation based on a mini-batch of samples to underestimate $q(z)$, $q(w)$ and $q(w_k)$.”
> 2. Since the practical calculation of the density $q(z)$, $q(w)$, and $q(w_i)$ is not the credit of our paper, we also added the very detailed description of how to conduct the underestimation in Appendix B.3 in the revised version.
>
> [1] Ricky TQ Chen, Xuechen Li, Roger B Grosse, and David K Duvenaud. Isolating sources of disentanglement in variational autoencoders. In Advances in Neural Information Processing Systems, pp. 2610–2620, 2018.
> ***
> **Comment #9:** *p6: you say you use the normalized mutual information between y_k and w_k. Please explain how you define this and how you calculate it in practice.*
>
> **Response #9:**
> 1. Practically, we utilize the classic tool “sklearn” with its function  “normalized_mutual_info_score”[2] to calculate the mutual information between $y_k$ and $w_k$. The input of the function is a set of property $y_k$ of the testing samples and a set of encoded latent variable $w_k$ of the testing samples. We have added this explanation in Footnote 1 on Page 7 in the revised version.
> 2. As defined officially in the function, Normalized Mutual Information (NMI) is a normalization of the Mutual Information (MI) score to scale the results between 0 (no mutual information) and 1 (perfect correlation).
>
> [2]https://scikitlearn.org/stable/modules/generated/sklearn.metrics.normalized_mutual_info_score.html#sklearn.metrics.normalized_mutual_info_score
> ***
> **Comment #10:** *p7, fig3 "... when traversing three latent variables in subset z ...". There are only 3 latent variables z? If more, how did you decide which shall be traversed?*
>
> **Response #10:** (1). For dSpirites dataset, there are 3 latent variables in $z$ and 3 latent variables in $w$. We used more latent variables in other datasets such as 100 latent variables in QM9 dataset.
> (2). When more latent variables are in $z$, we could have other ways to evaluate whether $z$ does not capture the information of properties, for example, calculating the avgMI as shown in Table 1.
> ***
> **Comment #11:**  *p7,p8, tab3: How do you predict the property from a molecule? You first infer w and z and then use the mean of $p(y | w)$ as prediction? Or you sample y from $p(y | w)$? Please elaborate.*
>
> **Response #11:**  Yes, you are right. We first infer $w$  from the input molecule $x$ by the inference model, and then use the invertible function of $p(y|w)$ to predict $y$. The output of the invertible function is the mean of $p(y|w)$ and is used as the prediction result. We have added this description to the second paragraph of Section 4.4 in the revised version. Thanks.
> ***

---

> ### Author Response · Authors · 2020-11-14
> **Responses to Reviewer4 (Part III)**
>
> **Comment #12:** *p7,p7, tab2: Do I read correctly that MSE of cLogP is bigger from the PCVAE (corr) then from the standard PCVAE? What does this say about the advantages of PCVAE (corr)?*
>
> **Response #12** Thanks for this comment.
> 1. Yes, you are right, the MSE of cLogP by PCVAE (corr) (i.e., 1.33) is a little larger than that of PCVAE (i.e. 1.29).
> 2. This is because PCVAE (corr) can deal with more general experiment setting than PCVAE, namely, PCVAE (corr) is designed to handle the situation where there are dependencies among properties: For example, the additional property clogS is correlated with cLogP. The results of PCVAE(corr) is still satisfying since it successfully captures the information of the added property cLogS with ignorable influence on the prediction of cLogP and Molweight.
> 3. We have added more analysis in the second paragraph in Section 4.4 in the revised version.
> ***
> **Comment #13:** *experiments over QM9: are all your evaluation metrics calculated only over valid molecules? Usually, generative models for molecules are evaluated using the validity, unicity and novelty scores. It would be instructive to complement your results with these.*
>
> **Response #13** (1) The evaluation metrics are calculated on all the generated molecules and the molecule generation model (CGVAE) we utilized has 100% validity. (2) We have added these evaluation results on validity, uniqueness, and novelty. Due to the space limit, we have added these evaluation results in Appendix C.1.
> ***
> **Minor Comment #1:** “Please provide references to models displayed in Fig1(a)-(c)”.
>
> **Response #1:** We have added the references in the caption of Figure 1 in the revised version.
> ***
> **Minor Comment #2:** *You use the term correletad a lot. I'm guessing here you mean any sort of dependence, not just linear?*
>
> **Minor Response #2:** Yes, you are right. In statistics, correlation or dependence is any statistical relationship (not limited to linear), whether causal or not, between two random variables [3].
>
> [3] "Correlation (in statistics)", Encyclopedia of Mathematics, EMS Press, 2001 [1994]
> ***
> **Minor Comment #3** *p2: "Directly enforcing such mutual independence inherently between all pairs of latent variables incurs exponential number of pairs among properties and latent variables for such enforcement." Exponential number of pairs? What do you mean?*
>
> **Minor Response #3:** Please refer to Response #5 to Comment #5 for this question. We have rephrased this in the Introduction part.
> ***
> **Minor Comment #4:** *p2, para3: "... as these have been shown to be relatively resilient with respect to the complex variants involved (Bengio et al., 2013)." Complex variants? What do you mean?*
>
> **Minor Response #4:**
> 1. This statement is concluded from the work by Bengio et al., 2013 [4], and borrowed from the sentence by Ma et al. [5] (we have added the reference [5] in the revised version).
> 2. The complex variants can be interpreted as a complex interaction of the explainable factors of complex data. As explained by Bengio et al, “Complex data arise from the rich interaction of many sources. These factors interact in a complex web that can complicate AI-related tasks such as object classification. For example, an image is composed of the interaction between one or more light sources, the object shapes, and the material properties of the various surfaces present in the image. Shadows from objects in the scene can fall on each other in complex patterns, creating the illusion of object boundaries where there are none and dramatically affect the perceived object shape.”
>
> [4] Bengio, Y., Courville, A., & Vincent, P. (2013). Representation learning: A review and new perspectives. IEEE transactions on pattern analysis and machine intelligence, 35(8), 1798-1828.
>
> [5] Ma, J., Cui, P., Kuang, K., Wang, X., & Zhu, W. (2019, May). Disentangled graph convolutional networks. In International Conference on Machine Learning (pp. 4212-4221).
> ***
> **Minor Comment #5：**  *p3, para2: $y=yk∈R_{k=1}^K$. Is this the same as $y∈R^K$?*
>
> **Minor Response #5:**
> 1. No. We are using $y=${$y_k∈R$}$_{k=1}^K$ in the paper, not $y=y_k∈R_{k=1}^K$.
> 2. $y$ is a set of properties and each property is denoted as $y_k$. $y_k∈R$ means that each property can be represented by a natural number. $y=${$y_k∈R$}$_{k=1}^K$ refers to that there are $K$ properties in the set $y$.
> ***
> **Typos errors**:
>
> **Response:** Thanks for pointing these. We have modified all the typos in the revised paper.

---

### Author Response · Authors · 2020-11-14
**A final summary of updates in the paper**

We appreciate so much for reviewers' comments and feedbacks that made our paper further improved while we were addressing their concerns. We are also glad that reviewers approved our clarifications and are satisfied with how we addressed their comments.
The following provides a final summary of the updates:

1. We have provided the codes of this paper in the supplemental materials.
2. We added the experiment on a new dataset: 3Dshapes.
3. We added the ablation study experiment for spectral normalization.
4. We added the ablation study experiment regarding handling the correlated properties in Appendix C.4.
5. We have modified the typos and rephrased some hard-parsing sentences.
6. We have added the references in the caption of Figure 1.
7. We have added the clarification about invertible dependence in the first line after Eq (9) on Page 5.
8. We have added a related work analysis in the second paragraph in Section 2 on Page 2.
9. We added the detailed derivation of Eq (2) on Page 4 as well as in Appendix A.3 and A.4.
10. We have added the annotation of expectation in Eq (4) and detailed derivation of Eq (4) in Appendix A.1.
11. We have added a detailed description of the calculation of the density $q(z)$ in Appendix B.3.
12. We have added the evaluation results on validity, unicity, and novelty of QM9 dataset in Appendix C.1.
13. We have added a simple variation of our method in handling discrete property in Appendix A2.
14. We have added a detailed description of avgMI in Footnote 1 on Page 7 and in Appendix B.4.

---

### Decision · Program_Chairs · 2021-01-07
**Final Decision**

**Decision:**

Accept (Poster)

**Comment:**

This paper proposes a new approach to learning deep generative models with induced structure in the latent representation. All four reviewers gave the same score of 6 to this paper, showing a consensus that the paper is above the bar for acceptance. The authors did a commendable job of detailed replies to reviewer comments, which as R1, R3, and R4 all note has improved the clarity and quality of the paper, addressing their concerns.